# Monte Carlo Tree Diffusion for System 2 Planning

**Jaesik Yoon** [1][2]  **Hyeonseo Cho** [1]  **Doojin Baek** [1]  **Yoshua Bengio** [3]  **Sungjin Ahn** [1][4]

## Abstract

Diffusion models have recently emerged as a powerful tool for planning. However, unlike Monte Carlo Tree Search (MCTS)—whose performance naturally improves with inference-time scaling—standard diffusion-based planners offer only limited avenues for scalability. In this paper, we introduce Monte Carlo Tree Diffusion (MCTD), a novel framework that integrates the generative strength of diffusion models with the adaptive search capabilities of MCTS. Our method reconceptualizes the denoising as a tree-structured process, allowing partially denoised plans to be iteratively evaluated, pruned, and refined. By selectively expanding promising trajectories while retaining the flexibility to revisit and improve suboptimal branches, MCTD achieves the benefits of MCTS such as controlling exploration-exploitation trade-offs within the diffusion framework. Empirical results on challenging long-horizon tasks show that MCTD outperforms diffusion baselines, yielding higher-quality solutions as inference-time computation increases.

## 1. Introduction

Diffusion models have recently emerged as a powerful approach to planning, enabling the generation of complex trajectories by modeling trajectory distributions using large-scale offline data (Janner et al., 2022; Ajay et al., 2023; Zhou et al., 2024; Chen et al., 2024a;c;b). Unlike traditional autoregressive planning methods, diffusion-based planners, such as Diffuser (Janner et al., 2022), generate entire trajectories holistically through a series of denoising steps, eliminating the need for a forward dynamics model. This approach effectively addresses key limitations of forward models, such as poor long-term dependency modeling and

error accumulation (Hafner et al., 2022; Hamed et al., 2024), making it particularly well-suited for planning tasks with long horizons or sparse rewards.

Despite their strengths, it remains uncertain how diffusion-based planners can effectively enhance planning accuracy through the computation scaling on the inference time—a property referred to as inference-time scalability. One potential approach is to increase the number of denoising steps or, alternatively, draw additional samples (Zhou et al., 2024). However, it is known that performance gains from increasing denoising steps plateau quickly (Karras et al., 2022; Song et al., 2021a;b), and independent random searches with multiple samples are highly inefficient as they fail to leverage information from other samples. Moreover, how to effectively manage the exploration-exploitation tradeoff within this framework also remains unclear.

In contrast, Monte Carlo Tree Search (MCTS) (Coulom, 2007), a widely adopted planning method, demonstrates robust inference-time scalability. By leveraging iterative simulations, MCTS refines decisions and adapts based on exploratory feedback, making it highly effective in improving planning accuracy as more computation is allocated. This capability has established MCTS as a cornerstone in many System 2 reasoning tasks, such as mathematical problem-solving (Guan et al., 2025; Zhang et al., 2024a) and program synthesis (Brandfonbrener et al., 2024). However, unlike diffusion-based planners, traditional MCTS relies on a forward model for tree rollouts, inheriting its limitations including losing global consistency. In addition to being restricted to discrete action spaces, the resulting search tree can grow excessively large in both depth and width. This leads to significant computational demands, particularly in scenarios involving long horizons and large action spaces.

This raises a crucial question: *how can we combine the strengths of Diffuser and MCTS to overcome their limitations and enhance the inference-time scalability of diffusion-based planning?* To address this, we propose Monte Carlo Tree Diffusion (MCTD), a framework that integrates diffusion-based trajectory generation with the iterative search capabilities of MCTS for more efficient and scalable planning.

MCTD builds on three key innovations. First, it restructures denoising into a tree-based rollout process, enabling semi-

---

[1]KAIST [2]SAP [3]Mila [4]New York University. Correspondence to: Jaesik Yoon <jaesik.yoon@kaist.ac.kr>, Sungjin Ahn <sungjin.ahn@kaist.ac.kr>.

*Proceedings of the 42$^{nd}$ International Conference on Machine Learning*, Vancouver, Canada. PMLR 267, 2025. Copyright 2025 by the author(s).

autoregressive causal planning while maintaining trajectory coherence. Second, it introduces guidance levels as meta-actions to dynamically balance exploration and exploitation, ensuring adaptive and scalable trajectory refinement within the diffusion framework. Third, it employs fast jumpy denoising as a simulation mechanism, efficiently estimating trajectory quality without costly forward model rollouts. These innovations enable the four steps of MCTS (Selection, Expansion, Simulation, and Backpropagation) within diffusion planning, effectively bridging structured search with generative modeling. Experimental results show that MCTD outperforms existing approaches in long-horizon tasks, achieving superior scalability and solution quality.

The main contributions of this paper are as follows: First, to the best of our knowledge, this is the first work to propose an MCTS-integrated diffusion planning framework that explicitly incorporates the four steps of MCTS, providing an effective inference-time scaling method for diffusion models. Second, we introduce three key innovations: Denoising as Tree-Rollout, Guidance Levels as Meta-Actions, and Jumpy Denoising as Fast Simulation. Lastly, we present experimental results demonstrating the effectiveness of MCTD.

## 2. Preliminaries

### 2.1. Diffuser: Diffusion Models for Planning

Diffuser (Janner et al., 2022) addresses long-horizon decision-making by treating entire trajectories as a matrix

$$\mathbf{x} = \begin{bmatrix} s_0 & s_1 & \dots & s_T \\ a_0 & a_1 & \dots & a_T \end{bmatrix},$$

where $s_t$ and $a_t$ denote the state and action at time t, respectively. A *diffusion process* is then trained to iteratively remove noise from samples of $\mathbf{x}$, ultimately producing coherent trajectories. In practice, this corresponds to reversing a forward noise-injection procedure by learning a denoiser $p_\theta(\mathbf{x})$ over the trajectory space. Since $p_\theta$ by itself does not encode reward or other task objectives, Diffuser optionally incorporates a heuristic or learned guidance function $J_\phi(\mathbf{x})$. This function predicts the return or value of a partially denoised trajectory, thereby biasing the sampling distribution:

$$\tilde{p}_\theta(\mathbf{x}) \quad \propto \quad p_\theta(\mathbf{x}) \exp\big(J_\phi(\mathbf{x})\big). \qquad (1)$$

Accordingly, at each denoising step, gradient information from $J_\phi$ nudges the model toward trajectories that appear both feasible (as learned from the offline data) and promising with respect to returns, in a manner akin to classifier guidance in image diffusion (Dhariwal & Nichol, 2021).

**Diffusion Forcing** (Chen et al., 2024a) extends the above framework by allowing tokenization of $\mathbf{x}$. This tokenization allows each token to be denoised at a different noise level, enabling partial denoising of segments where uncertainty

is high (e.g., future plans), without requiring a complete transition from full noise to no noise across the entire trajectory. Such token-level control is particularly beneficial in domains that demand causal consistency, such as long-horizon planning problems.

### 2.2. Monte Carlo Tree Search

Monte Carlo Tree Search (MCTS) is a planning algorithm that combines tree search with stochastic simulations to effectively balance exploration and exploitation (Coulom, 2007). Typically, MCTS proceeds in four stages: *selection*, *expansion*, *simulation*, and *backpropagation*. During *selection*, the algorithm starts at a root node corresponding to the current state (or partial plan) and descends the tree according to a policy such as Upper Confidence Bounds for Trees (UCT) (Kocsis & Szepesvári, 2006), aiming to choose the most promising child at each step or explore unexperienced states for balanced. Once a leaf or expandable node is reached, *expansion* adds new child nodes representing previously unexplored actions or plans.

Following expansion, *simulation* estimates the value of a newly added node by sampling a sequence of actions until reaching a terminal condition or a predefined rollout depth. The resulting outcome (e.g., cumulative reward in a Markov Decision Process) then updates the node's estimated value through *backpropagation*, propagating this information to all ancestor nodes in the tree. Over multiple iterations, MCTS prunes unpromising branches and refines its value estimates for promising subtrees, guiding the search toward more effective solutions.

## 3. Monte Carlo Tree Diffusion

In this section, we first outline the key concepts that enable MCTD planning: (1) Denoising as Tree-Rollout, (2) Guidance Levels as Meta-Actions, and (3) Jumpy Denoising as Simulation. These innovations form the foundation of MCTD, bridging the gap between traditional tree search methods and diffusion-based planning. We then describe the MCTD process in terms of the four steps: Selection, Expansion, Simulation, and Backpropagation.

### 3.1. Denoising as Tree-Rollout

Traditional MCTS operates on individual states, leading to deep search trees and significant scalability challenges. Because each node in the tree represents a single state, the depth of the tree increases linearly with the planning horizon, resulting in an exponential growth of the search space. Furthermore, it lacks the holistic perspective of the entire trajectory that diffusion-based planners inherently provide. On the other hand, Diffuser does not offer the tree-like struc-

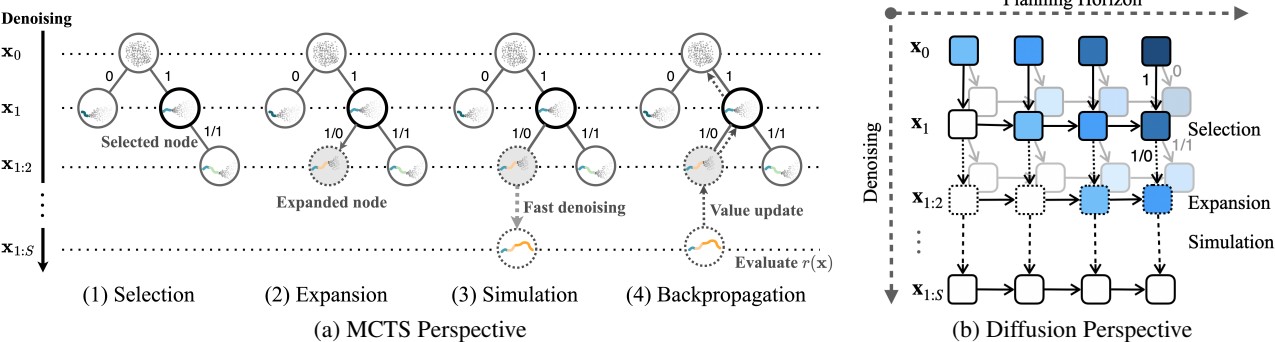

*Figure 1.* Two perspectives on Monte Carlo Tree Diffusion (MCTD). **(a) MCTS Perspective:** The four steps of an MCTD round—*Selection*, *Expansion*, *Simulation*, and *Backpropagation*—are illustrated on a partial denoising tree. Each node corresponds to a partially denoised sub-trajectory, and edges are labeled with binary guidance levels (0 = no guidance, 1 = guided). After a new node is expanded, "jumpy" denoising is performed to quickly estimate its value, which is then backpropagated along the path in the tree. **(b) Diffusion Perspective:** The same process is viewed as partial denoising across both denoising depth (vertical axis) and planning horizon (horizontal axis). Each colored block represents a partially denoised plan at a specific noise level, with darker shades indicating higher noise. Different expansions (0 or 1) create branches in the forward planning direction, representing alternative trajectory refinements. Notably, the entire row is denoised simultaneously, but with varying denoising levels. The MCTD framework unifies these two perspectives.

ture necessary for intermediate decision-point searches that effectively balance exploitation and exploration.

To address this issue, we first introduce the *Denoising as Tree-Rollout* process by leveraging the semi-autoregressive denoising process (Chen et al., 2024a). Specifically, we partition the full trajectory $\mathbf{x} = (x_1, x_2, \ldots, x_N)$ into $S$ subplans, $\mathbf{x} = (\mathbf{x}_1, \mathbf{x}_2, \ldots, \mathbf{x}_S)$ such that $\cap_s \mathbf{x}_s = \emptyset$. Unlike the standard diffuser, where all subplans share the same global denoising schedule, our approach allows assigning independent denoising schedules to each subplan. By applying faster denoising to earlier subplans and slower denoising to later subplans, the process effectively becomes causal and semi-autoregressive. This allows the denoising process to determine the future conditioned on the already determined past. Consequently, while *preserving the globally consistent and holistic generation advantages of Diffuser*, the denoising process approximates:

$$p(\mathbf{x}) \approx \prod_{s=1}^{S} p(\mathbf{x}_s | \mathbf{x}_{1:s-1}). \qquad (2)$$

Notably, although this formulation appears autoregressive, it is still executed as a single denoising process by controlling noise levels across subplans (Chen et al., 2024a).

In MCTD, each subplan $\mathbf{x}_s$, which represents a *temporally extended state*, is treated as a node in the search tree, rather than using individual states $x_n$ as nodes. This allows the tree to operate at a higher level of abstraction, improving efficiency and scalability. Denoising the entire plan $\mathbf{x}$ can be viewed as rolling out a sequence of these nodes through a single denoising process of Diffuser. Because $S \ll N$, the tree depth becomes much smaller than that in MCTS. For

instance, in one of our planning experiments, we use $S = 5$ and $N = 500$.

### 3.2. Guidance Levels as Meta-Actions

In MCTS, constructing and searching the tree becomes computationally expensive in large action spaces and is fundamentally impractical for continuous action spaces. To resolve this issue, in MCTD we introduce a novel approach by redefining the exploration-exploitation trade-off in terms of *meta-actions*. While meta-actions can be any discrete decision that can control the exploration-exploitation, in this work we propose to implement it as *guidance levels* applied during the denoising process.

For simplicity, consider two guidance levels: GUIDE and NO_GUIDE. In MCTD, we observe that sampling from the prior distribution $p(\mathbf{x})$, i.e., using the standard diffusion sampler represents exploratory behavior as it does not attempt to achieve any goal but only imitates the prior behavior contained in the offline data. Thus, we correspond it to the NO_GUIDE meta-action. Conversely, sampling from a goal-directed distribution $p_g(\mathbf{x})$, such as through classifier-guided diffusion (Dhariwal & Nichol, 2021), represents exploitative behavior, and thus meta-action, GUIDE, is assigned. This steers the sampling process toward achieving a specific goal defined by a reward function $r_g(\mathbf{x})$.

Next, we integrate the concept of meta-actions with the tree-rollout denoising process described in Eqn. (2). To achieve this, we introduce a guidance schedule $\mathbf{g} = (g_1, \ldots, g_S)$, which assigns a guidance meta-action $g_s \in \{\text{GUIDE}, \text{NO\_GUIDE}\}$ to each corresponding subplan $\mathbf{x}_s$. If no guidance is chosen, the subplan is sampled from the

explorative tree-rollout prior $p(\mathbf{x}_s|\mathbf{x}_{1:s-1})$ while a subplan with guidance is sampled from $p_g(\mathbf{x}_s|\mathbf{x}_{1:s-1})$ if the GUIDE meta-action is chosen. This allows the guidance levels to be independently assigned to each subplan unlike the standard Diffuser assigning the guidance throughout both the whole trajectory and the whole denoising process.

By dynamically adjusting the guidance schedule $\mathbf{g}$, this enables the exploration-exploitation balancing at the level of subplans *within a single denoising process*. The extended tree-rollout denoising process effectively approximates:

$$p(\mathbf{x}|\mathbf{g}) \approx \prod_{s=1}^{S} p(\mathbf{x}_s|\mathbf{x}_{1:s-1}, g_s). \tag{3}$$

As a result, this approach enables efficient and scalable planning, even in complex or continuous action spaces.

### 3.3. Jumpy Denoising as Fast Simulation

In MCTS, evaluating a node far from a leaf node, where evaluation of the plan is feasible, is a critical requirement. This is typically addressed in one of two ways: using a fast forward dynamics model to roll out trajectories to the leaf node, which is computationally expensive, or approximating the node's value via bootstrapping, offering faster but less accurate results. However, how to effectively incorporate these simulation strategies into the diffuser framework remains an open question.

In MCTD, we implement this simulation functionality using the *fast jumpy denoising* process based on the Denoising Diffusion Implicit Model (DDIM) (Song et al., 2021a). Specifically, when the tree-rollout denoising process has progressed up to the $s$-th subplan, the remaining steps are denoised quickly by skipping every $C$ step:

$$\tilde{\mathbf{x}}_{s+1:S} \sim p(\mathbf{x}_{s+1:S}|\mathbf{x}_{1:s}, \mathbf{g}). \tag{4}$$

This produces a full trajectory $\tilde{\mathbf{x}} = (\mathbf{x}_{1:s}, \tilde{\mathbf{x}}_{s+1:S})$, which is then evaluated using the reward function $r(\tilde{\mathbf{x}})$. While this fast denoising process may introduce larger approximation errors, it is highly computationally efficient, making it well-suited for the simulation step in MCTD.

### 3.4. The Four Steps of an MCTD Round

Building on the description above, we detail how the four traditional steps of MCTS—Selection, Expansion, Simulation, and Backpropagation—are adapted and implemented in MCTD. This process is illustrated in Figure 1.

**Selection.** In MCTD, the selection process involves traversing the tree from the root node to a leaf or partially expanded node. At each step, a child node is chosen based on a selection criterion such as Upper Confidence Bound (UCB). Importantly, this step does not require computationally expensive denoising; it simply traverses the existing

---

**Algorithm 1** Monte Carlo Tree Diffusion

```
 1: procedure MCTD(root, iterations)
 2:     for i = 1 to iterations do
 3:         node ← root
 4:         while ISFULLYEXPANDED(node) and not
 5:               ISLEAF(node) do
 6:             node ← BESTUCTCHILD(node)
 7:         end while
 8:         if ISEXPANDABLE(node) then
 9:             g_s ← SELECTMETAACTION(node)
10:             newSubplan ← DENOISESUBPLAN(node, g_s)
11:             child ← CREATENODE(newSubplan)
12:             ADDCHILD(node, child)
13:             node ← child
14:         end if
15:         partial ← GETPARTIALTRAJECTORY(node)
16:         remaining ← FASTDENOISING(partial)
17:         fullPlan ← (partial, remaining)
18:         reward ← EVALUATEPLAN(fullPlan)
19:         while node ≠ null do
20:             node.visitCount ← node.visitCount + 1
21:             node.value ← node.value + reward
22:             UPDATEMETAACTIONSCHEDULE(
23:                 node, reward)
24:             node ← node.parent
25:         end while
26:     end for
27:     return BESTCHILD(root)
28: end procedure
```

---

tree structure. Unlike traditional MCTS, MCTD nodes correspond to temporally extended states, enabling higher-level reasoning and reducing tree depth, which improves scalability. The guidance schedule $\mathbf{g}$ is dynamically adjusted during this step by UCB to balance exploration (NO_GUIDE) and exploitation (GUIDE).

**Expansion.** Once a leaf or partially expanded node is selected, the expansion step generates new child nodes by extending the current partially denoised trajectory. Each child node corresponds to a new subplan generated using the diffusion model. Depending on the meta-action $g_s$, the subplan is either sampled from the exploratory prior distribution $p(\mathbf{x}_s|\mathbf{x}_{1:s-1})$ or the goal-seeking distribution $p_g(\mathbf{x}_s|\mathbf{x}_{1:s-1})$. The meta-action controls which behavior is used for expansion, and the newly generated node is added to the tree as an extension of the current trajectory. Importantly, the guidance levels are not restricted to binary choices. For instance, it is possible to generalize the meta-actions to multiple guidance levels, such as {ZERO, LOW, MEDIUM, HIGH}, offering finer control over the balance between exploration and exploitation during the expansion process.

**Simulation.** Simulation in MCTD is implemented via the fast jumpy denoising. When a node is expanded, the remainder of the trajectory is quickly completed by using the fast denoising as described in Section 3.3. The resulting plan $\tilde{\mathbf{x}}$

is then given to the plan evaluator $r(\tilde{\mathbf{x}})$. This approach maintains computational efficiency while providing a sufficient estimate of the plan's quality.

**Backpropagation.** After the simulation step, the reward obtained from evaluating the complete plan is backpropagated through the tree to update the value estimates of all parent nodes along the path to the root. In MCTD, this backpropagation process also updates the meta-action-based guidance schedules, enabling the tree to dynamically adjust the exploration-exploitation balance for future iterations. This ensures that promising trajectories, as indicated by their rewards, are prioritized, while sufficient exploration is maintained to prevent premature convergence.

## 4. Related Works

Diffusion models (Sohl-Dickstein et al., 2015) have recently shown significant promise for long-horizon trajectory planning, particularly in settings with sparse rewards, by learning to generate entire sequences rather than stepwise actions (Janner et al., 2022; Chen et al., 2024a; Ajay et al., 2023; Zhou et al., 2024; Chen et al., 2024c;b; 2025). Various enhancements have been proposed to extend their capabilities. For instance, Chen et al.(2024a) incorporate causal noise scheduling for semi-autoregressive planning, and other works introduce hierarchical structures (Chen et al., 2024c;b; 2025), low-level value learning policies (Chen et al., 2024b), or classifier-free guidance (Ajay et al., 2023; Zhou et al., 2024). Despite these developments, the explicit interplay between exploration and exploitation in diffusion sampling has received relatively little attention.

Before Diffusion models, the long-horizon planning problem has been also addressed through the sub-goal based tree search (Jurgenson et al., 2020) and environment knowledge based high-level tree search (de Waard et al., 2016).

Chen et al. (2024a) further propose Monte Carlo Guidance (MCG) to leverage multiple sub-plans and average their guidance signals, thereby biasing denoising toward higher-reward outcomes. While this can encourage planning toward an expected reward over multiple rollouts, it does not implement an explicit search mechanism. The detailed discussion is in Appendix C. Similarly, Ye et al. (2025) apply a discrete diffusion denoising process to tasks such as chess, implicitly modeling MCTS without explicit tree expansion.

MCTS (Coulom, 2007) has achieved impressive results across various decision-making problems, particularly when combined with learned policies or value networks (Silver et al., 2016; Schrittwieser et al., 2020). It has also been applied to System 2 reasoning in Large Language Models (LLMs) to enhance structured problem-solving (Xiang et al., 2025; Yao et al., 2024; Zhang et al., 2024b). However, to the best of our knowledge, this work is the first to integrate tree search with diffusion models for full trajectory generation, bridging structured search with generative planning.

## 5. Experiments

We evaluate the proposed approach, MCTD, on a suite of tasks from the Offline Goal-conditioned RL Benchmark (OGBench) (Park et al., 2025), which spans diverse domains such as maze navigation with multiple robot morphologies (e.g., point-mass or ant) and robot-arm manipulation. Our chosen tasks—point and antmaze navigation, multi-cube manipulation, and a newly introduced visual pointmaze—jointly assess a planner's ability to handle *long-horizon planning*, *sequential manipulation*, and *partial visual observability*. In the visual pointmaze, an agent perceives RGB image observations of the 3D environment, thereby testing each method's resilience to partial observability and the ability to handle image-based planning. Detailed experimental settings are provided in Appendix A.

### 5.1. Baselines

We compare MCTD against several baselines, each employing a distinct strategy for test-time computation. First, we include the standard single-shot **Diffuser** (Janner et al., 2022), which plans only at the start of an episode and executes that plan without further adjustments. Next, we consider two methods that allocate additional test-time compute differently. **Diffuser-Replanning** periodically replans at fixed intervals, allowing partial corrections during the episode. This enables us to gauge the benefits of iterative replanning without a full tree search. **Diffuser-Random Search** (Zhou et al., 2024) generates multiple random trajectories in parallel and selects the highest-scoring one according to the same reward used by MCTD. While this increases test-time sampling and applies multiple guidance levels for different samples, it lacks the systematic expansion and pruning of a tree-search algorithm. We also compare our method to **Diffusion Forcing** (Chen et al., 2024a), which introduces a causal denoising schedule that enables semi-autoregressive trajectory generation. This comparison helps isolate the benefits of semi-autoregressive denoising from our full tree-structured approach. Further implementation details about all baselines can be found in Appendix A.1.

### 5.2. Maze Navigation with Point-Mass and Ant Robots

We begin with the pointmaze and antmaze tasks from OG-Bench, featuring mazes of varying sizes (medium, large, and giant). The agent is rewarded for reaching a designated goal region, and the relevant dataset (*navigate*) comprises long-horizon trajectories that challenge each method's capacity for exploration. As in prior work (Janner et al., 2022), we employ a heuristic controller for the pointmaze environment, whereas the antmaze environment uses a value-learning pol-

*Table 1.* **Long-Horizon Maze Results.** Success rates (%) on pointmaze and antmaze environments with medium, large, and giant mazes for the *navigate* datasets. Each cell shows the mean ± standard deviation.

| Environment | Dataset | Diffuser | | | Diffusion Forcing | MCTD |
|---|---|---|---|---|---|---|
| | | Base | Replanning | Random Search | | |
| pointmaze | medium-navigate-v0 | $58 \pm 6$ | $60 \pm 0$ | $60 \pm 9$ | $65 \pm 16$ | $\mathbf{100 \pm 0}$ |
| | large-navigate-v0 | $44 \pm 8$ | $40 \pm 0$ | $34 \pm 13$ | $74 \pm 9$ | $\mathbf{98 \pm 6}$ |
| | giant-navigate-v0 | $0 \pm 0$ | $0 \pm 0$ | $4 \pm 8$ | $50 \pm 10$ | $\mathbf{100 \pm 0}$ |
| antmaze | medium-navigate-v0 | $36 \pm 15$ | $40 \pm 18$ | $48 \pm 10$ | $90 \pm 10$ | $\mathbf{100 \pm 0}$ |
| | large-navigate-v0 | $14 \pm 16$ | $26 \pm 13$ | $20 \pm 0$ | $57 \pm 6$ | $\mathbf{98 \pm 6}$ |
| | giant-navigate-v0 | $0 \pm 0$ | $0 \pm 0$ | $4 \pm 8$ | $24 \pm 12$ | $\mathbf{94 \pm 9}$ |

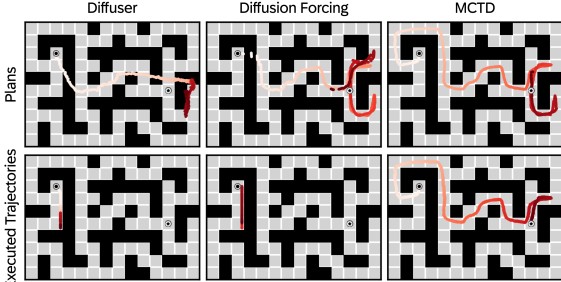

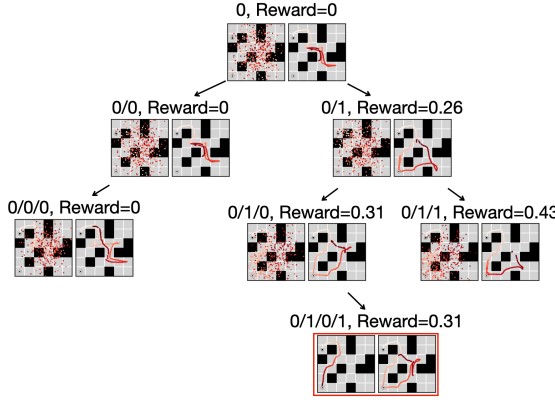

*Figure 2.* Comparison of generated plans and actual rollouts for three planners—Diffuser, Diffusion Forcing, and MCTD. While Diffuser and Diffusion Forcing fail to produce successful trajectory plans, MCTD succeeds by refining its plan adaptively.

icy (Chen et al., 2024b) for low-level control. Appendix A.5 provides more details on this task evaluation.

**Results** Table 1 presents success rates across medium, large, and giant mazes for both point-mass and ant robots. **MCTD consistently surpasses other methods by a large margin.** Notably, Diffuser-Random Search, despite using roughly the same number of denoising steps as MCTD, demonstrates no improvement over one-shot Diffuser, underscoring the importance of tree-based systematic branching over random sampling. Diffuser-Replanning similarly fails to outperform the base Diffuser, whereas Diffusion Forcing exhibits higher success rates in larger mazes, indicating benefits from a semi-autoregressive schedule for long horizons. However, only MCTD achieves near-perfect performance on the most challenging (large, giant) mazes, owing to its ability to refine partial trajectories and avoid dead-ends or inconsistent plan segments. This process is visualized in Figure 3. The performance on the giant map is impressive, because the map contains numerous detour paths to reach the goal as shown in Figure 2. The comparison with Diffuser-Random Search further highlights MCTD's efficiency, since it offers substantially better outcomes under the same computational budget.

In the antmaze domain, **MCTD continues to perform nearly perfectly, while baseline performance generally degrades more significantly than in pointmaze**, except for the medium-sized settings under Diffusion Forcing. A

*Figure 3.* Visualization of the MCTD tree-search process with a binary guidance set {NO_GUIDE, GUIDE} on a pointmaze-medium task. Each node corresponds to a partially denoised trajectory, where the left image shows the noisy partial plan and the right image shows the plan after fast denoising. The search expands child nodes by selecting either NO_GUIDE or GUIDE, evaluates each newly generated plan, and ultimately converges on the highlighted leaf as the solution.

principal reason for the degradation in the baselines is that the value learning policy (Wang et al., 2023) struggles in high-dimensional control tasks. Despite performance degradations due to inaccurate execution, MCTD consistently outperforms baselines and the performance gap widens as the maze size increases.

### 5.3. Robot Arm Cube Manipulation

For multi-cube manipulation tasks from OGBench, a robot arm must move one to four cubes to specific table locations as shown in Figure 4a. Increasing the number of cubes grows both the planning horizon and the combinatorial complexity. While the planner (MCTD or a baseline) issues high-level positions for the cubes, a value-learning policy (Wang et al., 2023) executes local actions. We augment MCTD with *object-wise guidance* by selecting which cube is manipulated at each step, avoiding simultaneous movements of multiple cubes. We note that the object-wise guidance is particularly suited to MCTD because the tree-expansion mechanism naturally supports discrete "meta-actions" at each node, such as selecting which object to move next.

*Table 2.* **Robot Arm Cube Manipulation Results.** Success rates (%) for single, double, triple, and quadruple cube tasks in OGBench. Parenthetical values in the MCTD-Replanning column denote performance when using a DQL performer trained only on the single-cube dataset.

| Dataset | Diffuser | Diffuser-Replanning | Diffusion Forcing | MCTD | MCTD-Replanning |
|---|---|---|---|---|---|
| single-play-v0 | $78 \pm 23$ | $92 \pm 13$ | $\mathbf{100 \pm 0}$ | $98 \pm 6$ | $\mathbf{100 \pm 0}$ |
| double-play-v0 | $12 \pm 10$ | $12 \pm 13$ | $18 \pm 11$ | $22 \pm 11$ | $50 \pm 16$ ($\mathbf{78 \pm 11}$) |
| triple-play-v0 | $8 \pm 10$ | $4 \pm 8$ | $16 \pm 8$ | $0 \pm 0$ | $6 \pm 9$ ($\mathbf{40 \pm 21}$) |
| quadruple-play-v0 | $0 \pm 0$ | $0 \pm 0$ | $0 \pm 0$ | $0 \pm 0$ | $0 \pm 0$ ($\mathbf{24 \pm 8}$) |

*Table 3.* **Visual Pointmaze Results.** Success rates (%) on partially observable, image-based mazes of medium and large sizes.

| Dataset | Diffuser | Diffuser-Replanning | Diffusion Forcing | MCTD | MCTD-Replanning |
|---|---|---|---|---|---|
| medium-navigate-v0 | $8 \pm 13$ | $8 \pm 10$ | $66 \pm 32$ | $82 \pm 18$ | $\mathbf{90 \pm 9}$ |
| large-navigate-v0 | $0 \pm 0$ | $0 \pm 0$ | $8 \pm 12$ | $0 \pm 0$ | $\mathbf{20 \pm 21}$ |

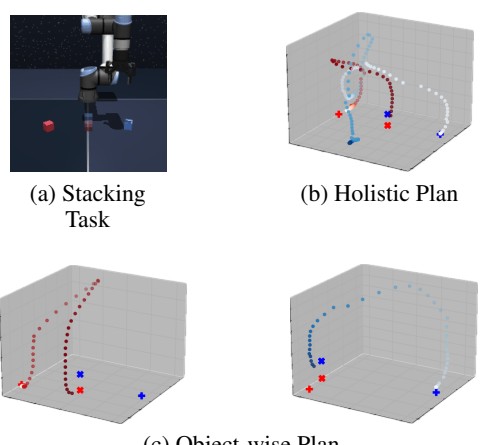

(a) Stacking Task
(b) Holistic Plan

(c) Object-wise Plan

*Figure 4.* (a) Illustration of the robot arm cube manipulation task. (b) A holistic plan for moving two cubes, where entanglement between their partial trajectories leads to an inaccurate movement (blue cube). (c) An object-wise plan produced by MCTD-Replanning, which avoids entanglement and achieves more reliable control over each cube's movement.

Single-pass diffusion methods, by contrast, attempt to produce a single, holistic trajectory in a single pass, leaving no opportunity to decide object-by-object. This guidance and additional task-specific reward shaping are described in Appendix A.6.

**Results**    Table 2 shows success rates for one to four cubes. All methods perform comparably on single-cube tasks, but performance drops significantly for multi-cube scenarios, especially for standard diffusion baselines that struggle to decide cube ordering and handle subtasks such as stacking. However, **MCTD-Replanning shows significant performance gaps from other models for multi-cube scenarios.** MCTD exhibits moderate gains (e.g., 22% on two cubes), though it suffers from holistic plan entanglements when multiple objects are involved as shown in Figure 4b. To address this, *MCTD-Replanning* periodically replans, effectively separating each cube's movements as shown in Figure 4c. This improves the success rate on the two-cube problem

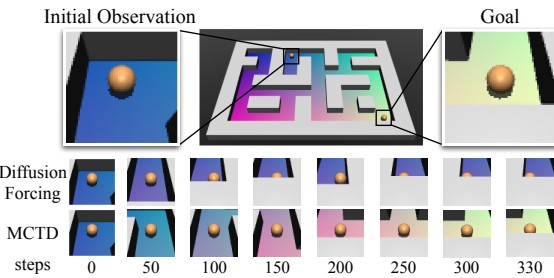

Initial Observation                    Goal

Diffusion Forcing

MCTD

steps        0    50    100    150    200    250    300    330

*Figure 5.* Task and interaction examples in the visual pointmaze. The agent operates under partial observability, receiving pixel-based initial and goal images. In the top row, Diffusion Forcing struggles to reach the goal due to a focus on exploitation, failing to sufficiently explore the environment. In contrast, MCTD (bottom row) effectively balances exploration and exploitation, successfully navigating to the goal.

from 22% to 50%, and also benefits tasks with more cubes. Another challenge arises from the value-learning policy, which generalizes poorly as the number of cubes increases. Even if MCTD generates a suitable global plan, flawed local control undermines the final outcome. Interestingly, training the policy solely on single-cube data still allows MCTD-Replanning to solve a portion (24%) of four-cube scenarios, illustrating the potential of object-wise guidance and iterative replanning.

### 5.4. Visual Pointmaze

To evaluate our method on image-based planning, particularly under partial observability, we introduce a visual pointmaze environment. In this environment, the agent receives top-down pixel images of the initial and goal states, as shown in Figure 5. Unlike the visual antmaze in OGBench, which involves complex locomotion, this environment allows us to focus on how well the planner operates with raw visual inputs without relying on ground-truth states.

We pre-encode observations with a Variational Autoencoder (VAE) (Kingma & Welling, 2014) to generate latent states suitable for diffusion-based planning, $z_t = \mathrm{VAE}(x_t)$. Because Diffuser's default heuristic controller is inapplica-

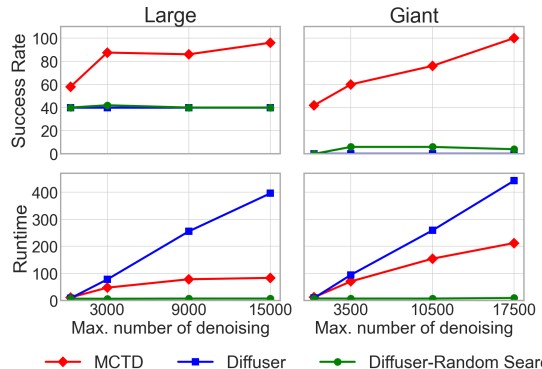

*Figure 6.* Success rates (%) and wall-clock runtime (Sec.) as the maximum number of denoising steps (test-time budget) increases for large and giant pointmaze tasks

ble to purely visual inputs, we design an *inverse dynamics* model that maps consecutive latent observations to actions, $\hat{a}_t = f_{\text{inv}}(z_{t-1}, z_t, z_{t+1})$. For guidance, we employ a position estimator trained on ground-truth coordinates; however, this estimator does not update the underlying VAE, ensuring that the planner remains in a partially observable regime. Further details on this task are provided in Appendix A.7.

**Results**   Table 3 compares success rates on medium and large versions of the visual pointmaze. **MCTD and MCTD-Replanning show better performance than baselines on image-based, partially observable tasks.** MCTD outperforms Diffuser baselines and Diffusion Forcing in the medium maze, presumably due to its tree-structured exploration, which better balances exploration and exploitation under perceptual partial observability. In the large maze, single-pass planners fail almost entirely, highlighting the difficulty of long-horizon visual planning. MCTD-Replanning improves performance by reevaluating partial plans midtrajectory, though success remains modest overall. This result underscores the inherent challenge of scaling diffusionbased methods to extended horizons with visual data under partial observability and suggests that future work may need more sophisticated visual encoders or hierarchical guidance strategies.

## 5.5. Inference-Time Scalability & Time Complexity

We examine how each planner leverages additional *test-time computation* by varying the maximum number of denoising steps and measuring both *success rate* and *runtime* on the large and giant pointmaze tasks (Figure 6). **As the inference-time budget grows, MCTD consistently achieves high success rates—ultimately nearing perfection in giant mazes.** In contrast, simply increasing denoising steps (Diffuser) or drawing more random samples (Diffuser-Random Search) yields modest returns, suggesting limited test-time scalability.

MCTD incurs additional overhead from maintaining and

expanding a tree—leading to a larger runtime as the budget grows. For instance, Diffuser-Random Search shows moderate runtime growth because multiple samples can be denoised in parallel batches, which is not directly feasible in MCTD's tree-structured approach. Nonetheless, MCTD can terminate early whenever a successful plan is found, thereby limiting unnecessary expansions. As a result, on the large maze, MCTD's runtime quickly saturates once the task can be solved with a smaller search depth, rather than fully consuming the available budget. A similar pattern emerges in the giant maze: as success rates increase under higher budgets, the runtime growth for MCTD remains sublinear, indicating that many expansions terminate early once suitable trajectories are located.

## 5.6. Ablation Studies

In this section, we conduct empirical ablation studies to analyze key components of MCTD, including the statistical nature of tree search, exploration-exploitation balance, tree depth, and the effectiveness of causal denoising and tree search. These ablation studies are primarily conducted on the PointMaze environment using the *navigate* datasets.

### 5.6.1. GREEDY TREE SEARCH

To evaluate the effectiveness of statistical tree search in balancing exploration and exploitation, we implemented **a greedy tree search baseline**. This approach searches candidates at each branching point and selects the best sample using the same reward function employed in Diffuser-Random Search and MCTD. For comprehensive evaluation, we varied the number of children for this approach from 5 to 20 (MCTD uses 5 children by default).

*Table 4.* Greedy Tree Search Performances (%) on PointMaze.

| Children# | 5 | 10 | 15 | 20 |
|---|---|---|---|---|
| Medium | 62±6 | 58±11 | 60±0 | 60±0 |
| Large | 40±0 | 40±0 | 40±0 | 40±0 |
| Giant | 0±0 | 0±0 | 0±0 | 0±0 |

The limited performance shown in Table 4 can be attributed to **the method's lack of deep search across multiple trajectory samples and the absence of backtracking mechanisms inherent in proper tree structures.**

### 5.6.2. META-ACTION SET INVESTIGATION

A core mechanism of MCTD is balancing exploration and exploitation to generate globally optimal plans. In this section, we study this balance by analyzing different metaaction set configurations. We evaluate five meta-action sets ranging from exploration-biased to exploitation-biased configurations. Set 1 is exploration-biased, consisting of small guidance levels $[0, 0.02, 0.05, 0.07, 0.1]$, while Set 2 repre-

sents our default configuration $[0, 0.1, 0.5, 1, 2]$. Sets 3 and 4 are increasingly exploitation-biased with $[0.5, 1, 2, 3, 4]$ and $[4, 5, 6, 7, 8]$, respectively. Finally, Set 5 includes extreme guidance levels $[0.1, 0.1, 1, 10, 100]$ to test whether MCTD can effectively select appropriate guidance levels while ignoring extreme values.

*Table 5.* Performances of Diverse Meta-Action Sets on PointMaze.

| Sets | Set 1 | Set 2 | Set 3 | Set 4 | Set 5 |
|---|---|---|---|---|---|
| Medium | 88±16 | 100±0 | 100±0 | 98±6 | 100±0 |
| Large | 44±15 | 98±6 | 90±0 | 80±0 | 100±0 |
| Giant | 54±23 | 100±0 | 100±0 | 74±18 | 100±0 |

The results in Table 5 demonstrate that performance degrades when meta-actions are heavily biased toward either exploration (Set 1) or exploitation (Set 4). Notably, MCTD maintains robust performance with balanced meta-action sets, even when some values are extreme outliers (Sets 2, 3, and 5). This indicates that **maintaining diverse guidance options is more critical** than their specific values, provided both exploration and exploitation are adequately represented.

### 5.6.3. SUBPLAN LENGTH

Another key configurable component in MCTD is tree depth, which can be controlled through the subplan count $S$. While $S$ is adaptively chosen based on the environment's episode length, in this section we investigate the impact of varying tree depths by manually controlling $S$.

*Table 6.* Empirical Results for Subplan Counts on PointMaze-Large.

| $S$ | 1 | 3 | 5 | 20 |
|---|---|---|---|---|
| Success Rate | 80±0 | 90±10 | 100±0 | 92±10 |
| Run Time (sec.) | 11±0 | 19±4 | 65±11 | 131±10 |
| Search # | 111±31 | 64±50 | 190±37 | 500±0 |

The results in Table 6 reveal **an important trade-off between computational efficiency and performance**. With longer subplans (smaller $S$), the search space is reduced, leading to faster execution times but potentially compromising performance quality. Conversely, very short subplans (e.g., $S = 20$) require more search iterations, which may exhaust the computational budget before finding optimal solutions.

### 5.6.4. CAUSAL DENOISING AND TREE SEARCH

To investigate the effectiveness of causal denoising and tree search, we compared four variants: Diffusion Forcing without Causal Denoising (DFwoCD), Diffusion Forcing (DF), MCTD without Causal Denoising (MCTDwoCD), and MCTD.

The results reveal that both components contribute substan-

*Table 7.* Performances of Causal Denoising and Tree Search Ablation on PointMaze.

| Method | DFwoCD | DF | MCTDwoCD | MCTD |
|---|---|---|---|---|
| Medium | 44±22 | 40±21 | 87±16 | 100±0 |
| Large | 50±14 | 55±20 | 78±6 | 98±6 |
| Giant | 8±10 | 34±16 | 18±11 | 100±0 |

tially to MCTD's performance. **Tree search consistently improves performance across all maze sizes**. Meanwhile, **causal denoising provides particularly substantial gains for long-horizon planning, with performance improvements most pronounced in the giant maze**.

## 6. Limitations & Discussion

While MCTD enhances inference-time scalability by integrating search-based planning with diffusion models, it remains computationally expensive due to its System 2-style deliberative reasoning. Determining when to engage in structured planning versus relying on fast System 1 (model-free) planning is an open challenge, as expensive tree search may not always be necessary. Adaptive compute allocation based on task complexity or uncertainty could improve efficiency.

Another limitation is inefficiency in large-scale search spaces, where evaluating multiple trajectory hypotheses remains computationally demanding despite using low-dimensional meta-actions. A promising direction is amortized search, where the system meta-learns from inference-time search to improve exploration efficiency over time. Instead of treating initial exploration as random, MCTD could incorporate inference-time learning mechanisms to refine its search dynamically. Additionally, self-supervised reward shaping could improve trajectory evaluation in sparse-reward settings. Optimizing parallelized denoising, differentiable tree search, and model-based rollouts could further enhance efficiency.

## 7. Conclusion

We introduced Monte Carlo Tree Diffusion (MCTD), a framework designed to combine the best of both worlds: the structured search of Monte Carlo Tree Search and the generative flexibility of diffusion planning to enhance the inference-time scalability of System 2 planning. MCTD leverages meta-actions for adaptive exploration-exploitation, tree-structured denoising for efficient diffusion-based expansion, and fast jumpy denoising for rapid simulation. Experimental results demonstrate that MCTD outperforms existing approaches in various planning tasks, achieving superior scalability and solution quality. Future work will explore adaptive compute allocation, learned meta-action selection, and reward shaping to further enhance performance, paving the way for more scalable and flexible System 2 planning.

## Acknowledgements

This research was supported by GRDC (Global Research Development Center) Cooperative Hub Program (RS-2024-00436165) and Brain Pool Plus Program (No. 2021H1D3A2A03103645) through the National Research Foundation of Korea (NRF) funded by the Ministry of Science and ICT.

## Impact Statement

This work introduces Monte Carlo Tree Diffusion (MCTD), a framework that integrates diffusion models with Monte Carlo Tree Search (MCTS) for scalable, long-horizon planning. As a general-purpose planning framework, MCTD itself does not pose direct risks; however, care must be taken when applying it to safety-critical domains, where decision-making impacts human well-being or critical infrastructure. Depending on its implementation, MCTD could influence high-stakes decisions in areas such as autonomous systems, healthcare, or finance, necessitating robust oversight and alignment with ethical guidelines. Additionally, its computational demands raise considerations around energy efficiency and sustainability, highlighting the need for responsible AI deployment.

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

# A. Experiment Details

## A.1. Baselines

We compare MCTD against several baselines, each highlighting different ways to leverage inference-time scaling on diffusion models or broader planning strategies:

- **Diffuser** (Janner et al., 2022): We use the standard single-shot Diffuser as our primary diffusion baseline. It generates a full trajectory by denoising from maximum to zero noise in one pass, guided by a return-predicting function $J_\phi$. While it can produce coherent plans, it lacks any mechanism to adapt or refine these plans at test time.

- **Diffuser-Replanning**: To partially address the lack of iterative search, we evaluated a replanning variant of Diffuser. After an initial denoised plan is sampled, the policy is periodically re-invoked at predefined intervals (e.g., every 10 time steps), using the agent's new state as the "start" for a new diffusion process. This strategy leverages additional test-time compute by attempting incremental corrections but does not share information across different replans.

- **Diffuser-Random Search**: We consider a random-search variant of Diffuser that generates multiple trajectories from random noise and selects the best candidate according to a heuristic or learned value function. This approach can be seen as a "Sample-Score-Rank" method proposed in (Zhou et al., 2024), increasing test-time compute by drawing more samples but lacking the systematic exploration or adaptation found in tree-search paradigms.

- **Diffusion Forcing**: Diffusion Forcing (Chen et al., 2024a) extends Diffuser by introducing a tokenized, causal noise schedule. Sub-plans of the trajectory can be denoised at different rates, allowing partial re-sampling of specific time segments without re-optimizing the entire plan. In our experiments, we evaluate Diffusion Forcing with the Transformer backbone provided in their official repository, which performed better in our preliminary tests. As the Diffusion Forcing baseline is inherently designed for replanning, we follow that evaluation setting.

## A.2. Heuristic Controller, Value-Learning Policy, and Inverse Dynamics Model

A long-standing challenge in diffusion-based planning is preserving global trajectory coherence while managing local, high-dimensional state-action control (Chen et al., 2024a;b). For example, PlanDQ (Chen et al., 2024b) couples a high-level diffusion planner with a learned low-level policy. Similarly, our approach focuses the diffusion planner on lower-dimensional, representative state information (e.g., object or agent positions), while delegating detailed action inference to a low-level action module (i.e., a heuristic controller, a value-learning policy, or an inverse dynamics model). The details of this integration are discussed in Algorithm 10. Such a hierarchical design allows MCTD and baseline planners to concentrate on broader strategic decisions while leaving the finer-grained execution details to this action module.

## A.3. Model Hyperparameters

For reproducibility, we detail the hyperparameters used in our experiments. These settings were selected based on prior work and empirical tuning to ensure stable training and evaluation. Nearly identical hyperparameters were applied consistently across all tasks, except where task-specific configurations were necessary, which are discussed in their respective sections.

### A.3.1. DIFFUSER

Our Diffuser implementation is based on the official repository (https://github.com/jannerm/diffuser). While retaining the core architecture and training procedure, we introduce the following modifications to align with our experimental setup:

- **Guidance Function:** Whereas the original Diffuser applied goal-inpainting guidance for point-maze tasks, we employ a distance-based reward function to ensure a fair comparison across all baselines.

- **Replanning Variant:** To evaluate the impact of inference-time iterative refinement, we implemented a replanning strategy that periodically re-samples trajectories based on the agent's updated state.

- **Random Search Variant:** To assess the effect of scaling inference-time computation, we implement the "Sample-Score-Rank" variant of Diffuser from (Zhou et al., 2024). Furthermore, to incorporate varied guidance, we assign a guidance scale to each sample, drawn uniformly from the set $\{0.01, 0.05, 0.1, 0.2, 0.3\}$.

*Table 8.* Diffuser Hyperparameters

| Hyperparameter | Value |
|---|---|
| Learning Rate | $2e - 4$ |
| EMA Decay | 0.995 |
| Precision in Training/Inference | 32 |
| Batch Size | 32 |
| Max Training Steps | 20000 |
| Planning Horizon | Different for each task, please check in the task sections |
| Open Loop Horizon | 50 for Diffuser-Replanning, otherwise planning horizon |
| Guidance Scale | 0.1 |
| Beta Schedule | Cosine |
| Diffusion Model Objective | $x_0$-prediction |
| U-Net Depth | 4 |
| Kernel Size | 5 |
| The Number of Channels | 32, 128, 256 |

### A.3.2. DIFFUSION FORCING

Our implementation of Diffusion Forcing is based on the officially released source code (`https://github.com/buoyancy99/diffusion-forcing`). The following modifications were made:

- **Replanning Frequency:** Since Diffusion Forcing natively supports inference-time replanning, we evaluated it under comparable conditions by applying periodic plan refinement, similar to our Diffuser-Replanning variant.

- **Transformer Backbone:** To maintain consistency across models, we utilized the Transformer-based backbone provided in the Diffusion Forcing repository.

### A.3.3. MONTE CARLO TREE DIFFUSION (MCTD)

Our MCTD model inherits most hyperparameters from Diffusion Forcing. However, we introduce specific modifications and add several new hyperparameters related to the tree search component.

### A.3.4. VALUE-LEARNING POLICY

To address the challenges of complex action spaces in specific tasks, we integrate a value-learning policy adapted from the PlanDQ source code (Chen et al., 2024b) (`https://github.com/changchencc/plandq`).

### A.4. The Evaluation Details

We followed the OGBench (Park et al., 2025) task configurations, 5 tasks per each environment. For each task, we evaluated 10 random seeds per each model, and reported the averaged success rates and their standard deviations.

### A.5. Maze Navigation with Point-Mass and Ant Robots

### A.5.1. MODIFICATIONS FROM OGBENCH

Although the original OGBench datasets include random noise in the start and goal positions, we eliminated this noise to isolate model performance from environmental randomness. This modification facilitates clearer comparisons across methods and ensures that performance differences can be attributed to planning capabilities rather than uncontrolled stochasticity. Additionally, we incorporate velocity into the state representation, which the original benchmark does not provide, to apply the heuristic controller for **Point-Maze** tasks implemented in (Janner et al., 2022). For the **Ant-Maze Giant** task, we found cases where well-generated plans failed due to inaccurate execution. To prevent this, we increased the episode length from 1000 to 1500.

*Table 9.* Diffusion Forcing Hyperparameters

| Hyperparameter | Value |
|---|---|
| Learning Rate | $5e-4$ |
| Weight Decay | $1e-4$ |
| Warmup Steps | 10000 |
| Precision in Training | 16-mixed |
| Precision in Inference | 32 |
| Batch Size | 1024 |
| Max Training Steps | 200005 |
| The Number of Frame Stack | 10 |
| Planning Horizon | Different for each task, please check in the task sections |
| Open Loop Horizon | 50 |
| Causal Mask | Not Used |
| Guidance Scale | 3 for medium mazes and 2 for other mazes |
| Scheduling Matrix | pyramid |
| Stabilization Level | 10 |
| Beta Schedule | Linear |
| Diffusion Model Objective | $x_0$-prediction |
| DDIM Sampling eta | 0.0 |
| Network Size | 128 |
| The Number of Layers | 12 |
| The Number of Attention Heads | 4 |
| The Feedforward Network Dimension | 512 |

### A.5.2. GUIDANCE FUNCTION

To plan long-horizon trajectories while maintaining the effectiveness to reach the goal as quickly as possible, we applied a guidance function to minimize the distance between each state and the goal, represented as $\sum_i ||x_i - g||_2$ (Chen et al., 2024a). We note that for a fair comparison, we applied the same guidance style across all baselines. For example, Diffuser originally used goal-inpainting guidance for Point-Maze, but in this evaluation, we used the distance-based guidance instead.

### A.5.3. GUIDANCE SCALE

We applied different guidance scales to MCTD for different tasks. For **Point-Maze Medium and Large**, a scale set of $[0, 0.1, 0.5, 1, 2]$ was used. For **Point-Maze Giant**, we applied $[0.5, 1, 2, 3, 4]$. For the **Ant-Maze** tasks, a set of $[0, 1, 2, 3, 4, 5]$ was used.

### A.5.4. PLANNING HORIZON

For the **Medium and Large mazes**, we applied a planning horizon of 500, and for the **Giant maze**, a horizon of 1000. The trajectory horizons in the original datasets are 1000 for Medium and Large, and 2000 for Giant. We used a shorter planning horizon than the dataset horizon to generate more data through the sliding window technique.

### A.5.5. HEURISTIC CONTROLLER AND VALUE-GUIDED POLICY

For **Point-Maze**, we adopted the heuristic controller designed in (Janner et al., 2022). For **Ant-Maze**, we used a value-guided policy (Wang et al., 2023), as was done in (Chen et al., 2024b). However, unlike (Chen et al., 2024b), we only provide the planned position 10 steps ahead as the subgoal.

### A.5.6. MCTD REWARD FUNCTION

We designed a heuristic reward function. One component of this function penalizes large, physically impossible position changes between consecutive states. Another component provides a reward for reaching the goal. Reaching the goal earlier

*Table 10.* MCTD Hyperparameters

| Hyperparameter | Value |
| --- | --- |
| Learning Rate | $5e-4$ |
| Weight Decay | $1e-4$ |
| Warmup Steps | 10000 |
| Precision in Training | 16-mixed |
| Precision in Inference | 32 |
| Batch Size | 1024 |
| Max Training Steps | 200005 |
| The Number of Frame Stack | 10 |
| Planning Horizon | Different for each task, please check in the task sections |
| **Open Loop Horizon** | Same to planning horizon for MCTD, 50 for MCTD-Replanning |
| Causal Mask | Not Used |
| Scheduling Matrix | pyramid |
| **The Maximum Number of Search** | 500 |
| **Guidance Set** | Different for each task, please check in task sections |
| **The number of Partial Denoising** | 20 |
| **The Jumpy Denoising Interval** | 10 |
| Stabilization Level | 10 |
| Beta Schedule | Linear |
| Diffusion Model Objective | $x_0$-prediction |
| DDIM Sampling eta | 0.0 |
| Network Size | 128 |
| The Number of Layers | 12 |
| The Number of Attention Heads | 4 |
| The Feedforward Network Dimension | 512 |

yields a larger reward, calculated as $r = (H - t)/H$, where $H$ is the total horizon length and $t$ is the current timestep.

### A.5.7. VALUE-LEARNING POLICY VARIANCE

For the AntMaze tasks, we employed a value-learning policy following (Chen et al., 2024b). We observed instances where plans were generated successfully, but the policy subsequently failed to execute them effectively. To account for this execution variance, we conducted three trials per task and seed combination for all AntMaze experiments, reporting the maximum success rate.

### A.5.8. SHORT-HORIZON (STITCH) RESULTS

To examine performance when only short-horizon trajectories are available during training, we evaluated our method on the stitch datasets, with results summarized in Table 12. The trajectory horizon in these datasets is significantly shorter than the number of steps required to reach the goal (e.g., a horizon of 100 steps in the dataset versus over 200 required steps for medium map tasks). Consequently, we applied replanning to Diffuser, Diffusion Forcing, and MCTD. In contrast to the long-horizon dataset experiments, the Diffuser demonstrates superior performance over Diffusion Forcing, suggesting that its implicit stitching capability is more effective. Meanwhile, MCTD outperforms the baselines on the medium-sized map due to its multi-sample exploration. However, its search depth was constrained by the limited length of trajectories in the training data, resulting in suboptimal performance on the large map.

### A.5.9. COMPARISON WITH GOAL-INPAINTING DIFFUSER

The Diffuser (Janner et al., 2022) typically employs a learned guidance function to bias generated trajectories toward high-return outcomes. For PointMaze tasks, however, an alternative approach—*goal-inpainting guidance*—was introduced (Janner et al., 2022), which improves single-pass planning by "imagining" an intermediate trajectory whose final state is the

*Table 11.* Value-Learning Policy Hyperparameters

| Hyperparameter | Value |
|---|---|
| Learning Rate | $3e-4$ |
| Learning Eta | 1.0 |
| Max Q Backup | False |
| Reward Tune | cql_antmaze |
| The Number of Training Epochs | 2000 |
| Gradient Clipping | 7.0 |
| Top-k | 1 |
| Target Steps | 10 |
| Randomness on Data Sampling (p) | 0.2 |

*Table 12.* **Short-Horizon (Stitch) Maze Results.** Success rates (%) on reduced-horizon *stitch* variants of the pointmaze environment.

| Dataset | Diffuser-Replanning | Diffusion Forcing | MCTD-Replanning |
|---|---|---|---|
| pointmaze-medium-stitch-v0 | $76 \pm 12$ | $53 \pm 16$ | $90 \pm 14$ |
| pointmaze-large-stitch-v0 | $34 \pm 16$ | $20 \pm 0$ | $20 \pm 0$ |

designated goal. In the context of Diffuser-Replanning, we adapt goal-inpainting by providing the experienced trajectory as contextual information rather than as a direct denoising target.

As shown in Table 13, goal-inpainting outperforms the standard Diffuser with heuristic guidance across medium, large, and giant pointmazes. By treating the goal as the terminal state to inpaint, the Diffuser model effectively samples trajectories from both ends, leveraging local convolutional mechanisms to reconcile the start and goal conditions. Although this approach yields marked performance gains, **MCTD** remains superior in long-horizon scenarios, outperforming both the base and replanning variants of the goal-inpainting Diffuser. The key distinction lies in MCTD's tree-structured partial denoising and adaptive branching; while goal-inpainting enhances single-pass generation, it lacks the iterative exploration-exploitation loop of an explicit tree search. Moreover, the advantage of two-ended trajectory imagination diminishes as the maze size increases (e.g., giant mazes). Consequently, MCTD uncovers robust solutions even where goal-inpainting shows diminishing returns, underscoring the benefits of tree-based refinement in complex, long-range planning tasks.

### A.6. Robot Arm Cube Manipulation

#### A.6.1. OBJECT-WISE GUIDANCE

Given that the robot arm can only manipulate a single cube at a time, we adopt an *object-wise guidance* mechanism in MCTD. Rather than applying a single, holistic diffusion schedule to all cubes simultaneously, MCTD treats the selection of which cube to move as a meta-action. This design partially mitigates issues arising from plans that attempt to move multiple cubes concurrently, although multi-object manipulation still requires careful sequencing to avoid suboptimal interleaving.

#### A.6.2. GUIDANCE FUNCTION

The guidance function is the same as that used for the maze tasks (Appendix A.5.2). For MCTD and MCTD-Replanning, the guidance is applied on an object-specific basis, aiming to reduce the distance between an object's current state and its corresponding goal state.

#### A.6.3. GUIDANCE SET

For the cube manipulation tasks, a guidance set of $[1, 2, 4]$ is applied for each object. For instance, in a task with two cubes, the total guidance set size is 6, comprising a set of three values for each of the two objects.

*Table 13.* **Comparison Results with Goal-Inpainting Diffuser.** Success rates (%) on pointmaze and antmaze environments with medium, large, and giant mazes for the *navigate* datasets.

| Environment | Dataset | Diffuser | | Goal-Inpainting Diffuer | | MCTD |
|---|---|---|---|---|---|---|
| | | Base | Replanning | Base | Replanning | |
| pointmaze | medium-navigate-v0 | $58 \pm 6$ | $60 \pm 0$ | $84 \pm 8$ | $80 \pm 9$ | $\mathbf{100 \pm 0}$ |
| | large-navigate-v0 | $44 \pm 8$ | $40 \pm 0$ | $94 \pm 9$ | $84 \pm 17$ | $\mathbf{98 \pm 6}$ |
| | giant-navigate-v0 | $0 \pm 0$ | $0 \pm 0$ | $30 \pm 16$ | $34 \pm 21$ | $\mathbf{100 \pm 0}$ |
| antmaze | medium-navigate-v0 | $36 \pm 15$ | $40 \pm 18$ | $100 \pm 0$ | $94 \pm 9$ | $\mathbf{100 \pm 0}$ |
| | large-navigate-v0 | $14 \pm 16$ | $26 \pm 13$ | $86 \pm 9$ | $66 \pm 13$ | $\mathbf{98 \pm 6}$ |
| | giant-navigate-v0 | $0 \pm 0$ | $0 \pm 0$ | $12 \pm 10$ | $20 \pm 0$ | $\mathbf{94 \pm 9}$ |

### A.6.4. REWARD FUNCTION

To avoid generating unrealistic plans, we augmented the reward function from the maze environment with additional constraints. A plan is deemed unrealistic and assigned a reward of 0 if it violates any of the following rules:

- **No Simultaneous Movements:** While minor noise in the plan is permissible, significant concurrent movement of multiple objects is forbidden.

- **Stable Final Placement:** A cube's final position must be stable, either on the floor or on top of another cube, not suspended in mid-air.

- **No Collisions:** A cube cannot be placed in a location already occupied by another cube.

- **Clearance for Manipulation:** A plan cannot move a cube if another cube is positioned on top of it.

### A.6.5. PLANNING HORIZON

The planning horizon was set to 200 for single-cube tasks, matching the episode length. For tasks involving multiple cubes, the horizon was extended to 500.

### A.6.6. VALUE-LEARNING POLICY

Similar to the antmaze tasks, we employed a value-learning policy (Wang et al., 2023). This policy is trained to achieve subgoals suggested by the planner, which correspond to planned states 10 steps into the future.

### A.6.7. HANDLING REDUNDANT OPERATIONS IN OBJECT-WISE PLANNING

To accelerate object-wise planning, we integrated common, repetitive operations into the framework. These operations include automatically dropping an object when it reaches its goal and pre-positioning the robot arm before executing the plan for each object. Embedding these routines directly mitigates inconsistencies, such as planning a move beyond the arm's reach or attempting to grasp an object when the gripper is already occupied.

### A.7. Visual Pointmaze

#### A.7.1. SETUP

Since diffusion-based planners typically operate on state-action representations, we first learn an *image encoder* using a Variational Autoencoder (VAE) (Kingma & Welling, 2014). The resulting latent states serve as the input to our diffusion model. Notably, these latent encodings do not contain explicit positional information; consequently, the planner must infer the underlying geometry to navigate the maze indirectly. We design an *inverse dynamics model*—a compact Multi-Layer Perceptron (MLP)—to map consecutive latent observations to actions. This model replaces the heuristic controller from the original Diffuser architecture, which is not directly applicable in this partially observable setting.

#### A.7.2. GUIDANCE

To guide the diffusion process, we introduce a *position estimator* trained on ground-truth coordinates. Although this estimator provides an approximate distance to the goal, its learning signal is not used to update the image encoder. The estimator does

not supply the planner with the full underlying state, thereby ensuring the image encoder processes only pixel-level data. This design preserves partial observability in the core planning procedure and highlights the open question of how best to incorporate task-relevant signals in visual domains. We apply the same guidance function as in the maze task described in Appendix A.5.2.

### A.7.3. GUIDANCE SCALES

For this task, we use a set of guidance scales $\lambda \in \{0, 0.1, 0.5, 1, 2\}$ for both MCTD and MCTD-Replanning.

### A.7.4. DATASET GENERATION AND PLANNING HORIZON

We follow the dataset generation script from OGBench (Park et al., 2025) for the PointMaze environment, setting trajectory horizons to 1000 for the Medium and Large mazes. To generate more training data, we use a reduced planning horizon of 500 for these mazes and employ a sliding-window technique, consistent with our earlier pointmaze experiments.

### A.7.5. VARIATIONAL AUTOENCODER

We pretrain a VAE (Kingma & Welling, 2014) to encode $64 \times 64 \times 3$ RGB observations into an $8$-dimensional latent space. This compresses the high-dimensional visual inputs into a structured representation suitable for downstream planning. The VAE models the latent variable $z$ as a Gaussian distribution with mean $\mu$ and standard deviation $\sigma$, using the reparameterization trick:

$$z = \mu + \sigma \cdot \epsilon, \quad \text{where} \quad \epsilon \sim \mathcal{N}(0, I). \tag{5}$$

We use the sampled latent embedding $z$ as the input representation for all downstream tasks.

### A.7.6. INVERSE DYNAMICS AND POSITION ESTIMATOR MODELS

We trained the inverse dynamics and position estimator models on an offline dataset of latent states encoded by the pretrained VAE. Both models are implemented as MLPs.

**Inverse Dynamics.** This model, $f_{\text{inv}}$, predicts the action $\hat{a}_t$ required to transition from state $z_t$ to the subsequent state $z_{t+1}$. To account for velocity in a partially observable context, the model is conditioned on both the previous and current latent states, $z_{t-1}$ and $z_t$:

$$\hat{a}_t = f_{\text{inv}}(z_{t-1}, z_t, z_{t+1}). \tag{6}$$

For the initial timestep ($t = 0$), where $z_{t-1}$ is unavailable, we set $z_{-1} = z_0$.

**Position Estimator.** The position estimator predicts the agent's $(x, y)$ coordinates from the latent state $z_t$. It provides a weak positional signal to aid the planner for guidance while maintaining the overall partial observability of the task.

### A.7.7. THE NECESSITY OF REPLANNING IN COMPLEX ENVIRONMENTS

In this section, we analyze the failure modes of diffusion-based planners that do not employ replanning, such as Diffusion Forcing, in the visual pointmaze task. As illustrated in Figure 7, the absence of iterative correction leads to trajectory collapse, revealing significant challenges in pixel-based planning. These findings underscore the necessity of replanning for generating robust trajectories in complex and partially observable environments.

## B. MCTD Architecture Implementation

The Monte Carlo Tree Diffusion (MCTD) framework relies on three key elements within its tree-search loop: sub-plan noise scheduling, causal semi-autoregressive denoising, and partial, jumpy denoising. First, rather than assigning a single noise schedule to the entire trajectory, each sub-trajectory segment receives its own noise-level schedule. This design allows different segments to transition from high to low noise at distinct rates, thereby enabling focused refinement of earlier parts of the plan while preserving flexibility in later segments.

Next, we adopt a *causal noise schedule* (Chen et al., 2024a), which renders the denoising process semi-autoregressive: earlier plan segments undergo more thorough denoising, while subsequent ones shift more gradually from higher to lower noise. This preserves the globally consistent trajectory generation characteristic of diffusion models but also ensures that

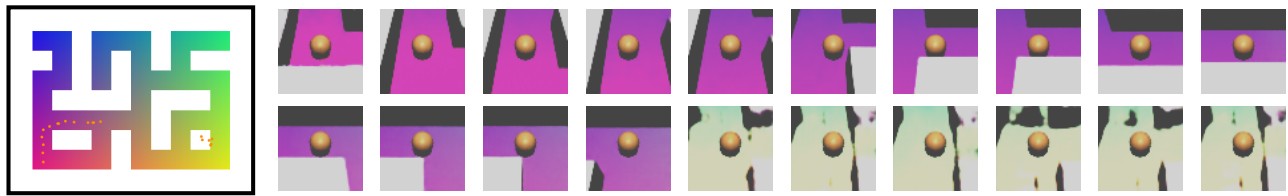

*Figure 7.* Trajectory collapse in Diffusion Forcing without replanning in the visual pointmaze. The left panel shows the estimated position of the planned trajectory (yellow points) halting at a certain area. The right panels (top left to bottom right) visualize the planning progression over time, suddenly teleporting to an unknown area This underscores the importance of replanning in overcoming single-pass planning limitations.

critical decisions near the beginning of the plan receive higher-fidelity refinements. Finally, both planning and simulation employ partial and jumpy denoising via DDIM sampling (Song et al., 2021a), allowing the system to skip directly from noise level $t$ to $t - \Delta$. This skip-level update accelerates the evaluation of candidate trajectories during tree expansion and supports partial denoising steps essential for exploration.

We implement these strategies with a Transformer-based model that processes an *expandable* token sequence, thus avoiding any architectural constraint on the planning horizon. Each sub-plan is annotated with a distinct noise-level index, and the Transformer predicts the fully denoised data $x_0$ from the corresponding noisy tokens. Concretely, we employ an $x_0$-parameterization by training the network to minimize the squared error between the predicted $\hat{x}_0$ and the ground-truth $x_0$:

$$\mathcal{L}_{\text{train}} = \mathbb{E}_{x_0, t, \epsilon}[\|\hat{x}_0 - x_0\|^2]. \tag{7}$$

Because each sub-plan has its own noise schedule, the Transformer naturally accommodates varying horizon lengths and partial denoising intervals without requiring further modifications to the model architecture.

By integrating causal scheduling with skip-level DDIM updates, MCTD achieves a semi-autoregressive, partial denoising process within a tree-search loop. Early plan segments are refined immediately, while future segments remain at higher noise until exploration or heuristic signals indicate their importance. This arrangement allows for fast approximate rollouts by skipping multiple noise levels when simulating a plan. Consequently, MCTD integrates the holistic generative capacity of diffusion models with the adaptive, branch-and-bound nature of Monte Carlo Tree Search, thereby enabling iterative trajectory refinement in tasks with long horizons and sparse rewards.

## C. Detailed Comparison between Monte Carlo Guidance (MCG) and Monte Carlo Tree Diffusion (MCTD)

While **MCG (Monte Carlo Guidance)** utilizes Monte Carlo sampling to adjust guidance levels during denoising, it lacks the structured search capabilities of **MCTD (Monte Carlo Tree Diffusion)** and does not explicitly refine trajectories over time. MCG focuses on adjusting guidance strengths rather than performing an explicit **trajectory search**, making it less effective for long-horizon planning where iterative improvement is crucial. Additionally, **MCG does not implement the four standard MCTS steps (Selection, Expansion, Simulation, and Backpropagation)**, limiting its ability to efficiently allocate inference-time scaling for progressive refinement. Without a structured search process, **MCG cannot revisit and optimize subplans**, making it less scalable for complex decision-making tasks. Furthermore, MCG lacks **meta-actions for adaptive exploration-exploitation**, passively adjusting guidance instead of dynamically balancing search strategies. Finally, it does not incorporate **fast jumpy denoising** for simulation, making trajectory evaluation less efficient compared to MCTD, which efficiently estimates plan quality without costly forward model rollouts.

# D. Additional Experiment Results

## D.1. UCT Hyperparameter Ablation Study

In addition to the meta-action ablation study (Section 5.6.2), we empirically analyze the exploration-exploitation balance by varying the UCT (Kocsis & Szepesvári, 2006) hyperparameter $W$ in the formula: $v_{\text{UCT}} = v_i + W\sqrt{\frac{\ln N}{n_i}}$, where $v_i$ is the node's estimated value, and $N$ and $n_i$ are the visit counts of the parent node and node itself, respectively.

Table 14. UCT Hyperparameter Ablation Study Results on PointMaze.

|  | $W$ | 0 (Greedy) | $\sqrt{2}$ (Default) | 3 | 5 | 10 |
|---|---|---|---|---|---|---|
| Medium | Success Rate (%) | 88±13 | 100±0 | 100±0 | 98±6 | 98±6 |
|  | Run Time (sec.) | 15±1 | 31±4 | 63±10 | 76±23 | 92±55 |
|  | The number of Search | 105±84 | 77±29 | 94±8 | 120±27 | 134±23 |
| Large | Success Rate (%) | 90±10 | 98±6 | 100±0 | 98±6 | 100±0 |
|  | Run Time (sec.) | 16±0 | 74±34 | 90±10 | 102±14 | 104±18 |
|  | The number of Search | 117±78 | 174±90 | 211±26 | 257±41 | 265±43 |
| Giant | Success Rate (%) | 82±14 | 100±0 | 98±6 | 100±0 | 100±0 |
|  | Run Time (sec.) | 25±1 | 215±23 | 216±12 | 225±22 | 230±10 |
|  | The number of Search | 228±69 | 394±152 | 442±13 | 464±13 | 493±7 |

As shown in Table 14, with $W = 0$ (pure greedy search), MCTD achieves faster inference times and requires fewer searches, but at the cost of reduced performance. This occurs because greedy search explores the tree depth-wise rapidly, reducing computational overhead of jump denoising but sacrificing thorough exploration. The default value ($W = 1.141$) strikes an effective balance, while higher values increase computational costs.

## D.2. Jumpiness Scale Ablation Study

Fast denoising (Song et al., 2021a) is a key concept in MCTD that enables tree search within the denoising process. The jumpiness scale $C$ determines how many denoising steps are skipped during this process. Larger $C$ values result in fewer denoising steps, trading accuracy for computational speed. We investigated how different jumpiness scales affect performance and efficiency, including a one-shot variant that denoises trajectories in a single step. The results demonstrate that proper selection of $C$ significantly impacts both model performance and efficiency.

Table 15. Jumpiness Scale Ablation Study Results on PointMaze.

|  | $C$ | 1 | 5 | 10 (Default) | 20 | 50 | One-shot |
|---|---|---|---|---|---|---|---|
| Medium | Success Rate (%) | 100±0 | 100±0 | 98±6 | 100±0 | 100±0 | 100±0 |
|  | Run Time (sec.) | 84±26 | 39±16 | 34±13 | 29±12 | 27±13 | 42±12 |
|  | The number of Search | 86±30 | 74±32 | 77±29 | 73±30 | 65±29 | 143±46 |
| Large | Success Rate (%) | 100±0 | 100±0 | 100±0 | 98±6 | 94±13 | 63±39 |
|  | Run Time (sec.) | 160±69 | 91±35 | 74±34 | 65±32 | 68±49 | 355±183 |
|  | The number of Search | 176±85 | 188±81 | 174±90 | 167±89 | 174±112 | 301±134 |
| Giant | Success Rate (%) | 100±0 | 98±6 | 100±0 | 100±0 | 88±16 | 10±21 |
|  | Run Time (sec.) | 632±239 | 258±115 | 231±103 | 185±88 | 174±76 | 164±2 |
|  | The number of Search | 390±148 | 379±158 | 394±152 | 363±154 | 387±149 | 500±0 |

For relatively simple tasks, even aggressive jumpiness (large $C$) or one-shot denoising maintains high performance. However, as task complexity increases, these approaches significantly degrade performance because value function estimation becomes less accurate under highly jumpy denoising, leading to suboptimal tree search decisions. The default setting ($C = 10$) achieves an excellent balance, maintaining high success rates while reducing computational time compared to smaller jump values. Additionally, we tested one-shot decoding by directly applying predictions from a diffusion model trained to predict fully denoised data. While this represents the most computationally efficient method, it exhibits less effective search behavior in medium-sized maps and clear performance degradation in larger maps. These results demonstrate that jumpy denoising provides an optimal balance between effectiveness and efficiency.

# E. Algorithms

---

**Algorithm 2** Monte Carlo Tree Diffusion (MCTD)

---

1: **procedure** MCTD($root, iterations$)
2:     **for** $i = 1$ to $iterations$ **do**
3:         $node \leftarrow$ SELECTPROMISINGNODE($root$)
4:         **if** ISEXPANDABLE($node$) **then**
5:             $node \leftarrow$ EXPANDNODE($node$)
6:         **end if**
7:         $reward \leftarrow$ SIMULATE($node$)
8:         BACKPROPAGATE($node, reward$)
9:     **end for**
10:     **return** BESTCHILD($root$)
11: **end procedure**

---

---

**Algorithm 3** Selection in MCTD

---

1: **procedure** SELECTPROMISINGNODE($node$)
2:     **while** ISFULLYEXPANDED($node$) **and not** ISLEAF($node$) **do**
3:         $node \leftarrow$ BESTUCTCHILD($node$)                         ▷ Use UCB for exploration-exploitation
4:     **end while**
5:     **return** $node$
6: **end procedure**

---

---

**Algorithm 4** Expansion in MCTD

---

1: **procedure** EXPANDNODE($node$)
2:     $g_s \leftarrow$ SELECTMETAACTION($node$)                      ▷ Determine guidance level
3:     $child \leftarrow$ DENOISESUBPLAN($node, g_s$)            ▷ Generate new subplan using diffusion
4:     ADDCHILD($node, child$)
5:     **return** $child$
6: **end procedure**

---

---

**Algorithm 5** Simulation in MCTD (Jumpy Denoising)

---

1: **procedure** SIMULATE($node$)
2:   $fullPlan \leftarrow$ FASTJUMPYDENOISING($node$)
3:   **return** EVALUATEPLAN($fullPlan$)
4: **end procedure**

---



---

**Algorithm 6** Backpropagation in MCTD

---

1: **procedure** BACKPROPAGATE($node, reward$)
2:   **while** $node \neq null$ **do**
3:    $node.visitCount \leftarrow node.visitCount + 1$
4:    $node.value \leftarrow node.value + reward$
5:    UPDATEMETAACTIONSCHEDULE($node, reward$)         ▷ Adjust guidance levels
6:    $node \leftarrow node.parent$
7:   **end while**
8: **end procedure**

---



---

**Algorithm 7** Meta-Action and Guidance Selection

---

1: **procedure** SELECTMETAACTION($node$)
2:   **return** UCBSELECTION($\{$NO, LOW, MEDIUM, HIGH$\}$)
3: **end procedure**
4: **procedure** DENOISESUBPLAN($node, g_s$)
5:   **if** $g_s =$ NO **then**
6:    **return** SAMPLEPRIOR($p(\mathbf{x}_s|\mathbf{x}_{1:s-1})$)           ▷ Exploration
7:   **else**
8:    **return** SAMPLEGUIDED($p_g(\mathbf{x}_s|\mathbf{x}_{1:s-1})$)         ▷ Exploitation
9:   **end if**
10: **end procedure**

---



---

**Algorithm 8** Jumpy Denoising for Fast Simulation

---

1: **procedure** FASTJUMPYDENOISING($partialPlan$)
2:   **return** SAMPLE($p(\mathbf{x}_{s+1:S}|\mathbf{x}_{1:s})$)           ▷ X-shot fast decoding
3: **end procedure**

---

---

**Algorithm 9** Partial and Jumpy Denoising with DDIM

---

**Require:** Initial plan $\mathbf{x}_{t_0}$ at noise level $t_0$, noise schedule $\mathcal{N}' = [t_0, t_1, \ldots, t_K]$, diffusion model $f_\theta(\cdot)$, guidance scale $g$
**Ensure:** Partially denoised plan $\mathbf{x}_{t_K}$

1: **for** $k = 0, 1, \ldots, K - 1$ **do**
  *// Predict noise or score at level $t_k$*
2: $\quad \boldsymbol{\epsilon} \leftarrow f_\theta\left(\mathbf{x}_{t_k}, t_k\right)$ $\qquad\qquad\qquad\qquad\qquad\qquad$ ▷ DDIM $\epsilon$-prediction or score function
  *// Apply guide using gradients of return*
3: $\quad \boldsymbol{\epsilon}^{(\text{guided})} \sim \mathcal{N}\left(\boldsymbol{\epsilon} + \alpha \Sigma \nabla \mathcal{J}, \Sigma^t\right)$
  *// DDIM update from $t_k$ to $t_{k+1}$*
4: $\quad \mathbf{x}_{t_{k+1}} \leftarrow \sqrt{\frac{\alpha_{t_{k+1}}}{\alpha_{t_k}}}\left(\mathbf{x}_{t_k} - \sqrt{1 - \alpha_{t_k}}\,\boldsymbol{\epsilon}^{(\text{guided})}\right) + \sqrt{1 - \alpha_{t_{k+1}}}\,\boldsymbol{\epsilon}^{(\text{guided})}$
5: **end for**
6: **return** $\mathbf{x}_{t_K}$

---

**Algorithm 10** Additional Controller/Policy/Inverse Dynamics Model Integration

---

**Require:** Long-term planner, $P_\theta$, environment $E$ and controller $\pi_\gamma$ (heuristic controller or value learning policy or inverse dynamics model), planning horizon $H$

1: $e \sim E$ with initial observation and goal $s, g$
2: **while not** *done* **do**
  *// Get a plan from planner $P_\theta$*
3: $\quad p_{1:H} = P_\theta(s, g)$
4: $\quad i = k$ $\qquad\qquad\qquad\qquad\qquad\qquad$ ▷ k steps after plan state is the subgoal for Controller
5: $\quad g' = p_k$
6: $\quad$ **while** $i \leq H$ **do**
    *// Compute low-level action with Controller*
7: $\quad\quad a \leftarrow \pi_\gamma(s, g')$
    *// Interact with environment*
8: $\quad\quad$ (o, r, done) $\leftarrow$ e.step(a)
9: $\quad\quad$ s $\leftarrow$ o
10: $\quad\quad$ **if** $s \approx g'$ **then**
11: $\quad\quad\quad i = i + k$
12: $\quad\quad\quad g' = p_k$
13: $\quad\quad$ **end if**
14: $\quad\quad$ **if** done **then**
15: $\quad\quad\quad$ break
16: $\quad\quad$ **end if**
17: $\quad$ **end while**
18: **end while**

---

