# OpenReview forum: "Monte Carlo Tree Diffusion for System 2 Planning"
_ICML.cc/2025/Conference — ICML 2025 spotlightposter_

### Official Review · Reviewer_6pZp · 2025-03-14

**Overall Recommendation:** 3

**Summary:**

The paper introduces Monte Carlo Tree Diffusion (MCTD) for long-horizon planning problems. MCTD combines the benefits of generative Diffusion models with tree-based planning in Monte Carlo Tree Search, without the requirement of a forward dynamics model. The primary aim is to utilise Diffusion models for improved long-horizon planning. Additionally, it allows MCTD to scale at test time with additional compute.

**Claims And Evidence:**

The authors make two claims.

One that MCTD outperforms standard Diffusion-based methods for long-horizon trajectory planning, which they demonstrate on three experiments (Maze Nav, Cube Manipulation, and Vizual Pointmase) and compare against various Diffusion-based baselines such as Diffuser (with various sampling regimes), and Diffusion Forcing.

Second, their application of MCTS with Diffusion models allows their Diffusion models to improve with additional Test Time Scaling which is demonstrated with one experiment on the Point Maze task.

**Essential References Not Discussed:**

The paper focuses on the Diffusion literature. It does not cite traditional planning literature that attempt to solve long-horizon planning problems via sub-plans [1], or Options [2]. Two suggestions would be:

[1] Jurgenson, T., Avner, O., Groshev, E. and Tamar, A., 2020, November. Sub-goal trees a framework for goal-based reinforcement learning. In International conference on machine learning (pp. 5020-5030). PMLR.

[2] M. de Waard, D. M. Roijers and S. C. J. Bakkes, "Monte Carlo Tree Search with options for general video game playing," 2016 IEEE Conference on Computational Intelligence and Games (CIG), Santorini, Greece, 2016, pp. 1-8,

**Experimental Designs Or Analyses:**

The experimental design and analysis appear sound and valid.

**Methods And Evaluation Criteria:**

The methods and evaluation criteria make sense for the problem introduced.

**Other Comments Or Suggestions:**

Typo: page 2 “aiming to balencely”

Page 4, should define DDIM.

**Other Strengths And Weaknesses:**

**Clarity**: A major strength of the work is the extremely well written manuscript. The Experiments and Results look promising, and the paper presents a new way of combining the benefits of Diffusion with long-horizon planning.

One weakness for me is the lack of details provided in the Experimental Setup. I don’t believe I could re-implement the proposed method from the manuscript alone. Whilst I believe most of the relevant information is provided in the Appendix (and code will be open-sourced), as a reader, I would benefit from a section in the main paper introducing topics such as: the Diffusion model architecture, the value networks, and training hyperparameters. Plus more details on the experiments themselves, like the action space, the training and compute budgets. This is all lacking in the main manuscript. Adding these details would improve the paper.

**Novelty and Significance**
To the best of my knowledge, this is the first application of Diffusion models to a tree-search framework such as MCTS, and the results are promising. The paper could be of interest to many researchers in the field.

**Questions For Authors:**

1) When partitioning trajectories into x_1, …, x_S, is S predetermined? How do you select it for each environment? and what effect does altering S have on the performance?

2) Since the Diffusion Forcing paper introduces a casual Denoising schedule, is it feasible to suggest that the additional performance gains are due to the tree-based planning? A more indepth discussion on this would be interesting.

3) What is C in the fast Denoising process? How does this affect the backpropagated values?

**Relation To Broader Scientific Literature:**

The paper situates itself relative to the Diffusion literature, and blends concepts from MCTS and tree-based planning to improve the behaviour of standard Diffusion models specifically for long-horizon planning.

**Theoretical Claims:**

None

---

> ### Author Rebuttal · Authors · 2025-04-01
>
> **Essential References Not Discussed**
>
> We thank you for recommending traditional long-horizon planning literatures. We will incorporate these valuable references in our revised version.
>
> > ... the lack of details provided in the Experimental Setup. ... lacking in the main manuscript.
> >
>
> Thank you for suggesting more experimental details in the main manuscript. In our revised version, we will include the experimental setup in the main text.
>
> > Typo: page 2 “aiming to balencely”, Page 4, should define DDIM.
> >
>
> Thank you for identifying these issues! We will correct them.
>
> > When partitioning trajectories into x_1, …, x_S, is S predetermined? How do you select it for each environment? and what effect does **altering S have on the performance**?
> >
>
> The subplan count $S$ is indeed a predetermined hyperparameter that we adapt to each environment's episode length. Following your insightful question, we conducted comprehensive ablation studies on PointMaze to analyze its effect. **The results demonstrate the suitable $S$ choice leads best effectiveness and efficiency.**
>
> **Subplan length ablation study**
>
> S=1 / 3 / 5 (Default) / 10 / 20
>
> Medium:
>
> - Performance: 80 ± 13 / 96 ± 8 / 100 ± 0 / 96 ± 8 / 98 ± 6
> - Run Time: 11 ± 1 / 14 ± 3 / 31 ± 4 / 103 ± 5 / 91 ± 3
> - Search#: 81 ± 63 / 29 ± 40 / 74 ± 9 / 475 ± 28 / 500 ± 0
>
> Large:
>
> - Performance: 80 ± 0 / 90 ± 10 / 100 ± 0 / 100 ± 0 / 92 ± 10
> - Run Time: 11 ± 0 / 19 ± 4 / 65 ± 11 / 118 ± 6 / 131 ± 10
> - Search#: 111 ± 31 / 64 ± 50 / 190 ± 37 / 500 ± 0 / 500 ± 0
>
> S=1 / 2 / 4 / 8 (Default) / 15 / 30
>
> Giant:
>
> - Performance: 72 ± 19 / 88 ± 16 / 100 ± 0 / 100 ± 0 / 100 ± 0 / 100 ± 0
> - Run Time: 13 ± 1 / 18 ± 4 / 40 ± 22 / 215 ± 23 / 213 ± 9 / 178 ± 7
> - Search#: 181 ± 78 / 75 ± 104 / 46 ± 41 / 374 ± 48 / 500 ± 0 / 500 ± 0
>
> These results reveal an important trade-off: With longer subplans (smaller S), the search space is reduced, leading to faster running times but potentially lower performance. Conversely, very short subplans (e.g., S=20) can require more search iterations, potentially exhausting the computational budget before finding optimal solutions.
>
> > Since the Diffusion Forcing paper introduces a casual Denoising schedule, is it feasible to suggest that **the additional performance gains are due to the tree-based planning?** A more indepth discussion on this would be interesting.
> >
>
> This is an excellent question about disentangling the contributions of tree search (TS) and causal denoising (CD). To address this directly, we conducted an ablation study comparing four configurations: Diffusion Forcing (DF) without CD, standard DF, MCTD without CD, and full MCTD. **The results demonstrate the huge performance gains from TS.**
>
> **Causal Denoising and Tree Search ablation studies**
>
> DF wo CD / DF / MCTD wo CD / MCTD Performances:
>
> - Medium: 44 ± 22 / 40 ± 21 / 87 ± 16 / 100 ± 0
> - Large: 50 ± 14 / 55 ± 20 / 78 ± 6 / 98 ± 6
> - Giant: 8 ± 10 / 34 ± 16 / 18 ± 11 / 100 ± 0
>
> The results reveal that both components contribute substantially to MCTD's performance. TS consistently improves performance across all maze sizes. Meanwhile, CD provides especially dramatic gains for long-horizon planning in the giant maze.
>
> > What is C in the fast Denoising process? How does **this affect the backpropagated values**?
> >
>
> The parameter $C$ in fast denoising determines how many levels are skipped during the jumpy denoising process. Larger $C$ values mean fewer denoising steps are performed, trading accuracy for speed. We investigated how different jumpiness scales affect performance and efficiency with the version which de-noises the trajectory in one-shot. **The results demonstrate proper value choice for $C$ affects the model performance and efficiency.**
>
> **Jumpiness scale ablation study**
>
> Jump=1 / 5 / 10 (Default) / 20 / 50 / One-shot
>
> Medium:
>
> - Performance: 100 ± 0 / 100 ± 0 / 98 ± 6 / 100 ± 0 / 100 ± 0 / 100 ± 0
> - Run time: 84 ± 26 / 39 ± 16 / 34 ± 13 / 29 ± 12 / 27 ± 13 / 42 ± 12
> - Search#: 86 ± 30 / 74 ± 32 / 77 ± 29 / 73 ± 30 / 65 ± 29 / 143 ± 46
>
> Large:
>
> - Performance: 100 ± 0 / 100 ± 0 / 100 ± 0 / 98 ± 6 / 94 ± 13 / 63 ± 39
> - Run time: 160 ± 69 / 91 ± 35 / 74 ± 34 / 65 ± 32 / 68 ± 49 / 355 ± 183
> - Search#: 176 ± 85 / 188 ± 81 / 174 ± 90 / 167 ± 89 / 174 ± 112 / 301 ± 134
>
> Giant:
>
> - Performance: 100 ± 0 / 98 ± 6 / 100 ± 0 / 100 ± 0 / 88 ± 16 / 10 ± 21
> - Run time: 632 ± 239 / 258 ± 115 / 231 ± 103 / 185 ± 88 / 174 ± 76 / 164 ± 2
> - Search#: 390 ± 148 / 379 ± 158 / 394 ± 152 / 363 ± 154 / 387 ± 149 / 500 ± 0
>
> For relatively simple tasks, even aggressive (large $C$) jumpiness or one-shot maintains high performance. However, as task complexity increases, they significantly degrades performance, as the value function estimation becomes less accurate under highly jumpy denoising, leading to suboptimal tree search decisions. The default (Jump=10) achieves an excellent balance, maintaining high success rates while reducing computational time compared to smaller jump.

---

### Official Review · Reviewer_TyXv · 2025-03-14

**Overall Recommendation:** 4

**Summary:**

The paper "Monte Carlo Tree Diffusion for System 2 Planning" introduces Monte Carlo Tree Diffusion (MCTD), a novel algorithm that merges diffusion models with Monte Carlo Tree Search (MCTS) to enhance test-time compute scalability in planning with diffusion models. It proposes three key innovations: restructuring denoising as a tree-rollout process for semi-autoregressive trajectory refinement, using guidance levels as meta-actions to balance exploration and exploitation, and employing jumpy denoising for planning. MCTD adapts the four MCTS steps—Selection, Expansion, Simulation, and Backpropagation—into a diffusion context, enabling iterative plan improvement. Evaluated on the Offline Goal-conditioned RL Benchmark, MCTD outperforms baselines like Diffuser and Diffusion Forcing in long-horizon tasks such as maze navigation , cube manipulation, and visual pointmaze. The paper claims MCTD is the first to explicitly integrate MCTS with diffusion planning, offering a scalable System 2 reasoning approach.

**After Rebuttal Summary**
The reviewers provided a comprehensive ablation that eased my main concerns that this was simply performing a form of standard sampling. I think this ablation improves the strength of the paper and the takeaways regarding the role that MCTS plays in performance.

**Claims And Evidence:**

Claims made are both supported and convincing.

**Essential References Not Discussed:**

No

**Experimental Designs Or Analyses:**

Yes sound.

**Methods And Evaluation Criteria:**

Evaluation datasets are appropriate.

**Other Comments Or Suggestions:**

No

**Other Strengths And Weaknesses:**

Strengths
- Novel integration of MCTS with diffusion that enables planning at a high level of abstraction, which is very promising for making test time planning methods more efficient.

Weaknesses
- Lack of ablations investigating the parameters of MCTS search. In particularly investigating the balancing of the exploitation-exploration trade off. This would give more insight as to whether the planning is more sampling based or really finding promising branches and exploiting them. Willing to raise my score if such ablations are added.

**Questions For Authors:**

- Could your method be more generally presented as planning at a higher level of abstraction and then open to a variety of planning algorithms, or is it inherently tied to an MCTS based approach?

- The readout policy used was best child in MCTS, did you explore any other readout policies?

**Relation To Broader Scientific Literature:**

The contributions of "Monte Carlo Tree Diffusion for System 2 Planning" bridge diffusion-based planning and Monte Carlo Tree Search (MCTS) literature. It builds on diffusion works like Diffuser (Janner et al., 2022) and Diffusion Forcing (Chen et al., 2024a) by adding tree-structured search to improve test-time compute scalability, addressing limitations noted in Karras et al. (2022), and extends guidance concepts from Dhariwal & Nichol (2021) into meta-actions. In the broader RL and MCTS context, it advances Coulom (2006) and applications like AlphaGo (Silver et al., 2016) by planning at a higher abstraction level with subplans as nodes, reducing reliance on error-prone forward models (Hafner et al., 2022) and aligning with hierarchical planning trends (Chen et al., 2024c), offering a novel, efficient alternative for long-horizon tasks and System 2 reasoning.

**Theoretical Claims:**

N/A

---

> ### Author Rebuttal · Authors · 2025-04-01
>
> **Relation To Broader Scientific Literatures**
>
> We thank you for acknowledging our work's connection to the broader scientific literatures. We will ensure these connections are clearly articulated in the revised version.
>
> > Lack of ablations investigating the parameters of MCTS search. In particularly investigating the **balancing of the exploitation-exploration trade off.**
> >
>
> We appreciate this valuable suggestion. Following your suggestion, ****we have conducted comprehensive ablation studies focusing on the MCTS hyperparameters that control exploration-exploitation balance in MCTD on PointMaze. **The results demonstrate the exploration-exploitation balance is one of the key factors to optimize the performance efficiently.** Below are key results from these studies:
>
> **Meta-action choice ablation study**
>
> Choice 1: [0, 0.02, 0.05, 0.07, 0.1]
>
> Choice 2: [0, 0.05, 0.1, 0.5, 1]
>
> Choice 3: [0, 0.1, 0.5, 1, 2]
>
> Choice 4: [0.5, 1, 2, 3, 4]
>
> Choice 5: [2, 3, 4, 5, 6]
>
> Choice 6: [4, 5, 6, 7, 8]
>
> Choice 7: [0, 0.1, 1, 10, 100]
>
> Choices: C1 / C2 / C3 / C4 / C5 / C6 / C7 Performances:
>
> - Medium: 88 ± 16 / 98 ± 6 / 100 ± 0 / 100 ± 0 / 98 ± 6 / 98 ± 6 / 100 ± 0
> - Large: 44 ± 15 / 100 ± 0 / 98 ± 6 / 90 ± 10 / 80 ± 0 / 80 ± 0 / 100 ± 0
> - Giant: 54 ± 23 / 100 ± 0 / 100 ± 0 / 100 ± 0 / 88 ± 16 / 74 ± 18 / 100 ± 0
>
> The results reveal that when meta-actions are heavily biased toward either exploration (C1) or exploitation (C6), performance degrades significantly. Interestingly, MCTD demonstrates robust performance with balanced meta-action sets, even when some values are extreme outliers (e.g., C7). This suggests that maintaining diverse guidance options is more important than their specific values, as long as both exploration and exploitation are adequately represented.
>
> **UCB hyperparameter ablation study**
>
> We investigated the influence of the UCB exploration weight parameter $C$ in the formula: $v_{\text{UCB}} = v_i + C\sqrt{\frac{\ln N}{n_i}}$, where $v_i$ is the node's estimated value, and $N$ and $n_i$ are the visit counts of the parent node and the node itself, respectively.
>
> C=0(Greedy) / 1.141 (Default) / 3 / 5 / 10
>
> Medium:
>
> - Performance: 88 ± 13 / 100 ± 0 / 100 ± 0 / 98 ± 6 / 98 ± 6
> - Run time: 15 ± 1 / 31 ± 4 / 63 ± 10 / 76 ± 23 / 92 ± 55
> - Search#: 105 ± 84 / 77 ± 29 / 94 ± 8 / 120 ± 27 / 134 ± 23
>
> Large:
>
> - Performance: 90 ± 10 / 98 ± 6 / 100 ± 0 / 98 ± 6 / 100 ± 0
> - Run time: 16 ± 0 / 74 ± 34 / 90 ± 10 / 102 ± 14 / 104 ± 18
> - Search#: 117 ± 78 / 174 ± 90 / 211 ± 26 / 257 ± 41 / 265 ± 43
>
> Giant:
>
> - Performance: 82 ± 14 / 100 ± 0 / 98 ± 6 / 100 ± 0 / 100 ± 0
> - Run time: 25 ± 1 / 215 ± 23 / 216 ± 12 / 225 ± 22 / 230 ± 10
> - Search#: 228 ± 69 / 394 ± 152 / 442 ± 14 / 464 ± 13 / 493 ± 7
>
> With C=0 (pure greedy search), MCTD shows faster inference time and conducts fewer searches, but performance degrades. This occurs because greedy search tends to explore the tree depth-wise quickly, reducing the overhead from jumpy denoising but compromising thorough exploration. The default value (C=1.141) achieves a good balance, while higher values show diminishing returns in performance despite increased computational costs.
>
> **Other ablation studies**
>
> We also studied other ablations for **subplan length**, **causal denoising**, **tree search**, and **the scale of jumpiness** which are discussed in the rebuttal for 6pZp's review.
>
> > Could your method be more **generally presented as planning** at a higher level of abstraction and then open to a variety of planning algorithms, or is it inherently tied to an MCTS based approach?
> >
>
> While we built our approach on an MCTS backbone, the core idea—treating partial denoising steps as higher-level "states" and using meta-actions (guidance choices) to expand them—is not inherently tied to MCTS. In principle, **one could substitute different search algorithms** (e.g., best-first search or A*) to explore and refine these partial trajectories. However, MCTS's ability to balance exploration-exploitation through principled tree search is uniquely suited for navigating the complex, high-dimensional trajectory spaces in diffusion planning.
>
> > The **readout policy** used was best child in MCTS, did you explore any other readout policies?
> >
>
> Thank you for suggesting this interesting idea. We agree that we may consider most visited child as an alternative. However, **it seems not be fitting well to our settings** as it pursues finding sequences of a complete high-quality plan instead of inferring the next best action. In future work, we plan to explore other readout strategies and their effects on planning in different domains.
>
> [1] Silver, David, et al. "Mastering the game of Go with deep neural networks and tree search." *nature* 529.7587 (2016): 484-489.

---

### Official Review · Reviewer_iXeu · 2025-03-15

**Overall Recommendation:** 4

**Summary:**

Standard diffusion-based planners don't improve with additional test-time computation. This limits their effectiveness on complex long-horizon tasks.The authors introduce Monte Carlo Tree Diffusion (MCTD) which formulates the diffusion process in such a way that each denoising process can effectively "branch out" - i.e. each denoising step can theoretically happen in different goal directions an with different denoising rates. Notably the "branching out" of diffusion step gives rise to a tree structure when different branches are explored. This in turn allows for it to search over these branches and build on the monte carlo tree search literature to efficiently explore different branches to find better overall plans at the cost of additional test-time computation.

The authors show that MCTD significantly outperforms standard diffusion planners and reinforcement learning baselines on challenging long-horizon tasks with sparse rewards albeit at the cost of large test-time computation.

## Post Rebuttal
I acknowledge the authors' rebuttal and maintain my original assessment.

**Claims And Evidence:**

Key Claim: MCTD formulation of diffusion allows to exploit test-time compute to improve the overall quality of long horizon plans.
Evidence: Quantitative results across maze navigation tasks show significant improvements over baselines.
Moreover the integration of MCTS principles into this diffusion framework is sound and the implementation details are detailed.

Minor comment 1: The claim of "System 2 planning" is bit of a stretch as the system does not implictly have a world model it can search all variations upon. At best its a very informed heuristic search over the data distribution and does not have any sense of planning in abstract space and grounding it to low level actions.

Minor comment 2:  The claim that MCTD "overcome limitations" of both MCTS and diffusion models is also a bit of a stretch. It is definitely the case that MCTD allows us to combine the theory of MCTS with the diffusion process. But it does it at the cost of test time compute and is a different dimension to doing the diffusion process and not really "overcoming limitations" in a pure sense. This does not "magically" solve the limitation of MCTS being slow and needing a model, or diffusion process being suboptimal when being used in its more standard forms.

**Essential References Not Discussed:**

NA

**Experimental Designs Or Analyses:**

- The choice of OGBench's maze environments provides a good testbed for long-horizon planning with sparse rewards
- Their ablation of "Diffuser with diffused action" highlights the failure mode of baseline approaches


concerns:
- Lack of direct comparison to other test-time optimization methods which can scale with compute. Random diffusion process can be easily scaled to any amount of compute for fair comparison. How about a baseline where the tree search is simply conducted greedily at every diffusion step? It could very well be the case that one step lookahead at every step would also have been enough.

- Missing analysis of how performance varies with subplan count and meta-action choices. IT would have been wonderful to see the effectiveness of the approach being highlighted for varying number of sub-plan count and meta action choices.

**Methods And Evaluation Criteria:**

The evaluation criteria is appropriate.
- The maze navigation tasks (point-mass and ant) provide clear tests of long-horizon planning capabilities with varying difficulty levels
- The multi-cube manipulation evaluates compositional planning and transfer in a physical domain.
- Visual maze navigation tests performance under partial observability
- The metrics (success rates and rewards) directly measure planning effectiveness

It is unfortunate that the random search using repeated diffusion was not scaled to match the test time compute used by MCTD. It would have been very useful to show that random search alone scales very poorly with more test time compute.

**Other Comments Or Suggestions:**

- Whle analogies are clear with MCTS, its is a bit hard to follow with references to "nodes" and "meta actions". May be it would have been simpler to simply define a diffusion MDP where each state is a set of partially denoised sub plans and action can denoise one subplan in different directions. This way may be it would have been easier to define and contrast directly with the MCTS formulation.

- It would be very interesting to reformulate the test time compute objective to have "diverse set of solutions" rather than just optimizing for the best solution.

**Other Strengths And Weaknesses:**

Weakneses:
- There could be simple but strong baselines that leverage test time compute in trivial ways to match the compute used by MCDS. For example, greedy search on the MCDS tree structure, repeated runs of random diffuser-random search.
- The work does not provide detailed study on the effects of sub-plan length and meta action choices in a controlled setting.
- The work also does not show what is the trade-off between using the scale of "jumpiness" in the jumpy denoising and overall plan quality.

**Questions For Authors:**

- Did the authors visualize a particular example where UCB style approach had clear benefits over just greedy search ?

**Relation To Broader Scientific Literature:**

Understanding the nuances of diffusion process and leveraging it for faster generation of plans has been explored by some related works where they directly predict a candidate first and then do a limited number of diffusion steps. This work explores the opposite direction where test time compute is being exploited.

leveraging test time compute has always been grounded in search in one form or another. Searching all different paths the diffusion process could have taken and exploring through them has close connections with tree of thought literature from other generative models like LLMs.

**Theoretical Claims:**

While relationship with MCTS has been established and is clearly motivated, theoretical results will be very hard to inherit as the "fast diffusion" does not use any model whose sub-optimality can be quantified. Moreover the action spaces defined in the diffusion process is technically continuous with a non trivial dynamics model which has not been studied for th eproblem.

---

> ### Author Rebuttal · Authors · 2025-04-01
>
> > The claim of "System 2 planning" is bit of a stretch as the system does not implictly have a world model it can search all variations upon.
> >
>
> We acknowledge that there may be differing perspectives on what constitutes System 2 planning in the context of ML. Drawing on Kahneman [1], we define System 2 planning as a deliberate, sequential search across multiple hypotheses at inference time, which aligns with our approach.
>
> Regarding world models, we argue that diffusion models can qualify. Although they may not strictly meet the action-conditioned forward model definition, we believe a narrow view of “world model” is unnecessary for System 2 classification. Since diffusion models implicitly encode structural knowledge, we question whether excluding them is truly warranted.
>
> Our method uses a semi-autoregressive rollout guided by meta-actions, similar to autoregressive, action-conditioned forward models but operating in a more temporally abstract space. We believe that this abstraction retains the hypothesis-driven nature of System 2 planning and supports our contribution to broader discussions on such systems.
>
> [1] Kahneman, Daniel, 1934-2024, author. Thinking, Fast and Slow. New York :Farrar, Straus and Giroux, 2011.
>
> > The claim that MCTD "overcome limitations" of both MCTS and diffusion models is also a bit of a stretch.
> >
>
> We acknowledge that our phrasing could be more precise. Rather than "overcoming limitations," MCTD brings complementary strengths from each approach while addressing certain weaknesses. We will revise this claim more moderately in the final version.
>
> > It is unfortunate that the random search using repeated diffusion was not scaled to **match the test time compute** used by MCTD.
> >
>
> Major computation overhead on Diffusion planner is on denoising process, and **we evaluated some baselines with matched budget in our paper.** Diffuser-Random Search baseline already uses the same number of denoising steps as MCTD, as discussed in Section 5.2. Additionally, Section 5.5 compares both Diffuser and Diffuser-Random Search against MCTD using equivalent computational budgets.
>
> > theoretical results will be very hard to inherit as the "fast diffusion" does not use any model whose sub-optimality can be quantified.
> >
>
> We agree on this. Thank you pointing out this!
>
> > How about a baseline where the tree search is simply conducted **greedily at every diffusion step**?
> >
>
> Following your recommendation, we implemented and evaluated one-step greedy search at every diffusion step on Diffuser for the PointMaze. The approach can search diverse candidates at each step and select the best using the same reward function as MCTD. We scaled this approach to use even more children than MCTD (MCTD uses 5 children). **The results demonstrate that TTC scaling is indeed limited with this approach.** Specifically, we did:
>
> Child# = 5 / 10 / 15 / 20
>
> Performances:
>
> - Medium: 62±6 / 58±11 / 60±0 / 60±0
> - Large: 40±0 / 40±0 / 40±0 / 40±0
> - Giant: 0±0 / 0±0 / 0±0 / 0±0
>
> The results are primarily due to the lack of deep searching across multiple samples and the absence of backtracking over tree structure.
>
> > Missing analysis of how **performance varies with subplan count** and **meta-action choices**.
> >
>
> Thanks to your feedback, we **additionally conducted comprehensive ablation studies for both subplan length and meta-action choices on PointMaze**. They are discussed in the rebuttal for reviewers 6pZp and TyXv due to the space limitation.
>
> > Searching all different paths the diffusion process could have taken and exploring through them has close **connections with tree of thought** literature from other generative models like LLMs.
> >
>
> Thank you for pointing out this. We agree that our work shares important connections with Tree of Thought approaches. We will strengthen these references in our revised paper.
>
> > There could be simple but strong baselines ... greedy search on the MCDS tree structure, **repeated runs of random diffuser-random search**.
> >
>
> We addressed this in the evaluation for one-step greedy search.
>
> > The work does not provide detailed study on the effects of sub-plan length and meta action choices in a controlled setting.
> >
>
> We addressed this in the performance analysis for subplan count and meta-action choices.
>
> > The work also does not show what is **the trade-off between using the scale of "jumpiness" in the jumpy denoising and overall plan quality**.
> >
>
> We appreciate this insightful suggestion. We **additionally investigated the effects of the "jumpiness" scale** through additional ablation study, which is discussed in the rebuttal for 6pZp's review.
>
> > Did the authors visualize a particular example where **UCB style approach had clear benefits** over just greedy search ?
> >
>
> We appreciate this insightful suggestion. We **additionally conducted an ablation study controlling the weight C for visit count in the UCB value formulation** which is discussed in the rebuttal for TyXv's review.

---

> > ### Comment · Reviewer_iXeu · 2025-04-04
> >
> > I confirm that I have read the author response to my review. I thank the authors for the added experiments in the rebuttal phase. This helps make a stronger case for the approach. I stand by my decision to accept the paper.

---

### Official Review · Reviewer_Fb4y · 2025-03-17

**Overall Recommendation:** 2

**Summary:**

This paper proposes a new algorithm, MCTD, which involves MCTS in the diffusion model. MCTD uses whole trajectories as its states and introduces a meta-action to generate child nodes based on whether a guidance function is used. By leveraging the MCTS process, MCTD achieves better success rates and runtime performance in most test settings compared to baselines such as Diffuser and Diffusion Forcing.

**Claims And Evidence:**

in detailed comments

**Essential References Not Discussed:**

NA

**Experimental Designs Or Analyses:**

in detailed comments

**Methods And Evaluation Criteria:**

in detailed comments

**Other Comments Or Suggestions:**

Code only has README, and README is empty.

Preliminaries

Diffuser equation 1: Why is there an exp which is different from the diffuser paper?



Monte Carlo Tree Diffusion

3.1

“Specifically, we partition the full trajectory… x_s!=empty.” What is x_i in x? What is S?

What is “subplan” and “denoising schedule”? I am totally lost in this paragraph.

“Because S ≪ N.” What is N?

I assume x_s is an agent path trajectory but with noise (middle result of the diffusion model). So basically, this part wants to say MCTD uses x_s as a tree node, so the tree height depends on the diffusion model iteration steps. If we only need 5 steps to recover the whole trajectory from noise, the tree height will be 5.

3.2

“While meta-actions… during the denoising process.” So what are “meta-actions”? I still have no idea about them.

“In MCTD, we observe… in the offline data.” I don’t understand, and there may be some grammar problems.

To my understanding, the meta-action is just two sample models: one samples from p, and the other samples from p_g. According to figure 1, it seems MCTD alternates between using two kinds of probabilities. In the tree, each node only has two child nodes. So the question is, could we have more child nodes, or how could we determine the optimal number of child nodes?

3.3

The author should ensure the notations are consistent or explain new notations such as g in equation 4 and g_s in equation 3.

My question is, if we already have a reward function and also a guidance function, why do we need a no_guidance action? I think, just like in the diffuser model, encoding this information in the inference part should be enough.



Experiments

Since the baselines are mainly diffuser models, I think there should be a test using the same maze map and robot stacking test settings as in the diffuser paper.

**Other Strengths And Weaknesses:**

Strengths:

The idea of MCTD is interesting.

MCTD performance surpasses baselines in most test cases, both in success rate and runtime.


Weaknesses:

The meta-action is not very convincing; I am not sure why we need a no-guidance action.

The writing should be largely improved.

**Questions For Authors:**

In the tree, each node only has two child nodes. So the question is, could we have more child nodes, or how could we determine the optimal number of child nodes?

If we already have a reward function and also a guidance function, why do we need a no_guidance action? I think, just like in the diffuser model, encoding this information in the inference part should be enough.

**Relation To Broader Scientific Literature:**

in detailed comments

**Theoretical Claims:**

NA

---

> ### Author Rebuttal · Authors · 2025-04-01
>
> > I am not sure why we need a **no-guidance action**.
> >
>
> Diffusion planners often rely on strong guidance, but under unseen goals or complex long-horizon tasks, this can be detrimental. Allowing “no-guidance” (or weak guidance) provides essential exploration. We demonstrate this by **showing the performance degradation when MCTD only uses high-guidance values.**
>
> Default (Guidance Levels: [0, 0.1, 0.5, 1, 2]) / Only High Guidance (Guidance Levels: [2, 2, 2, 2, 2]) Performances:
>
> - Medium: 100 ± 0 / 98 ± 6
> - Large: 98 ± 6 / 80 ± 0
> - Giant: 100 ± 0 / 78 ± 11
>
> > Code only has README, and README is empty.
> >
>
> We are working on the code release. We will make sure to release the code after the review process.
>
> > Why is there an exp which is different from the diffuser paper?
> >
>
> This reflects a probabilistic interpretation common in RL, where optimality can be linked to exponentiating the reward, $\exp(r(\tau))$. In the original Diffuser paper, a similar concept was denoted by a heuristic term $h(\tau)$. We write $\exp(\mathcal{J}(\tau))=h(\tau)$.
>
> > What is $\mathbf{x}_i$ in $\mathbf{x}$? What is $S$ ?
> >
>
> As written in the page 3, line 136, $\mathbf{x}_i$﻿ is subplan in full trajectory $\mathbf{x}$﻿, and $\bigcap_S \mathbf{x}_S=\emptyset$  means there is no shared trajectories between subplans. $S$ is the number of subplans in the full trajectory $\mathbf{x}$.
>
> > What is “subplan” and “denoising schedule”?
> >
>
> As written in the page 3, line 136, subplan is the part of trajectories. Denoising schedule is the assignments of different noise levels per each subplan. For instance, in MCTD, near future subplans are considered as low noised, while further future subplans are as highly noised as shown in Figure 1 (b).
>
> > “Because S ≪ N.” What is N?
> >
>
> N is the total length of trajectories and S is the number of subplans.
>
> > I assume x_s is an agent path trajectory but with noise (middle result of the diffusion model). ... so the tree height depends on the diffusion model iteration steps.
> >
>
> $\mathbf{x}_s$ denotes a **subplan** (a partial trajectory segment) at some intermediate stage of denoising, not the full trajectory. In MCTD, the **tree height is determined by the number of subplans** $S$, rather than the total denoising steps.
>
> > What are "meta-actions"?
> >
>
> In standard MCTS, each node expands by choosing from all possible actions. Because we want to keep the branching factor manageable (especially in continuous spaces), we introduce **meta-actions** to represent **guidance levels**.
>
> > “In MCTD, we observe… in the offline data.” I don’t understand, and there may be some grammar problems.
> >
>
> Although we checked that its grammar is not wrong, we agree that it could be clearer with a slight change as follows:
>
> *In MCTD, we observe that sampling from the prior distribution p(x), i.e., using the standard diffusion sampler, represents exploratory behavior as it does not attempt to achieve any goal. Instead, it only imitates the prior behavior contained in the offline data.*
>
> In the part, we discussed the sampling from the prior $p(\mathbf{x})$ represents the exploratory behavior by not attempting to achieve any goal.
>
> > could we have **more child nodes**, or how could we determine the **optimal number of child nodes**?
> >
>
> We show a simple illustration in the Figure 1 with two children. In practice, we can use **multiple guidance levels** and indeed we do in our experiments.
>
> Currently, the number of child nodes (meta-actions) is a **hyperparameter**. Exploring an adaptive selection of the number of meta-actions for each task is an interesting direction for future work.
>
> > The author should ensure the notations are consistent or explain new notations such as g in equation 4 and g_s in equation 3.
> >
>
> $\mathbf{g}$ and $g_s$ are introduced in page 3 as the guidance schedule and guidance, but we will revise to reintroduce them right before each equation.
>
> > why do we need a no_guidance action?
> >
>
> We replied in the no-guidance action investigation.
>
> > ... I think there should be a test using **the same maze map** and robot stacking test settings as in **the diffuser paper**.
> >
>
> We appreciate the suggestion. While we could not evaluate MCTD on the robot stacking due to time limitation, we **did** evaluate on the D4RL multi-2D maze tasks which are evaluated in Diffuser paper. **The results demonstrate MCTD outperforms Diffuser.**
>
> Diffuser / MCTD Performances:
>
> - Umaze: 128.9 ± 1.8 / 239.7 ± 46.2
> - Medium: 127.2 ± 3.4 / 430.6 ± 104.8
> - Large: 132.1 ± 5.8 / 548.9 ± 195.8
>
> We measured the performance with the accumulated rewards from the environment as done in Diffuser. The performance is averaged from 100 samples. We analyzed the performance gaps due to the searchablity on MCTD and different guidance types (Diffuser - inpainting / MCTD - reducing distances between states and goal). The inpainting guidance can lead inefficient planning when the start and goal are nearer than the plan length.

---

> > ### Comment · Reviewer_Fb4y · 2025-04-06
> >
> > Thank you for the response. I believe this paper could benefit from some revisions. My score remains primarily due to concerns about the writing issue.

---

### Decision · Program_Chairs · 2025-05-01

**Decision:**

Accept (spotlight poster)

**Comment:**

This paper combines Monte Carlo tree search (MCTS) with diffusion for the purpose of planning. By augmenting diffusion planning with MCTS, it is possible to leverage additional test time compute to improve the quality of the plans, a capability that vanilla diffusion doesn't have.

All reviewers with a substantial input agree that this is a strong paper that deserves publication. Although not a magic bullet (the increase in computational cost is significant), it shows a non-trivial way to leverage additional compute in diffusion planning, a topic of significant focus for the ICML community. Experiments are well-designed and convincingly show the improved efficiency, including long-horizon tasks, compositional planning, partial observability, etc. The rebuttal also shows the benefits of the proposed approach wrt one-step greedy search (a more trivial integration of MCTS and diffusion). Overall, convincing and relevant work with solid results.